# Multi-species single-cell transcriptomic analysis of ocular compartment regulons

Pradeep Gautam[1,2,14], Kiyofumi Hamashima[1,14], Ying Chen[1,2,3], Yingying Zeng[1,4], Bar Makovoz[5], Bhav Harshad Parikh[6,7], Hsin Yee Lee[1], Katherine Anne Lau[1], Xinyi Su [6,7,8], Raymond C. B. Wong [9,10,11], Woon-Khiong Chan [2,3], Hu Li [12✉], Timothy A. Blenkinsop [5✉] & Yuin-Han Loh [1,2,3,13✉]

The retina is a widely profiled tissue in multiple species by single-cell RNA sequencing studies. However, integrative research of the retina across species is lacking. Here, we construct the first single-cell atlas of the human and porcine ocular compartments and study inter-species differences in the retina. In addition to that, we identify putative adult stem cells present in the iris tissue. We also create a disease map of genes involved in eye disorders across compartments of the eye. Furthermore, we probe the regulons of different cell populations, which include transcription factors and receptor-ligand interactions and reveal unique directional signalling between ocular cell types. In addition, we study conservation of regulons across vertebrates and zebrafish to identify common core factors. Here, we show perturbation of *KLF7* gene expression during retinal ganglion cells differentiation and conclude that it plays a significant role in the maturation of retinal ganglion cells.

[1] Cell Fate Engineering and Therapeutics Laboratory, A*STAR Institute of Molecular and Cell Biology, Singapore 138673, Singapore. [2] Department of Biological Sciences, National University of Singapore, Singapore 117543, Singapore. [3] Integrative Sciences and Engineering Programme (ISEP), NUS Graduate School, National University of Singapore, 21 Lower Kent Ridge Road, Singapore 119077, Singapore. [4] School of Biological Sciences, Nanyang Technological University, Singapore 637551, Singapore. [5] Icahn School of Medicine at Mount Sinai, New York, NY 10029, USA. [6] Department of Ophthalmology, Yong Loo Lin School of Medicine, National University of Singapore, Singapore, Singapore. [7] Translational Retinal Research Laboratory, A*STAR Institute of Molecular and Cell Biology, Singapore 138673, Singapore. [8] Singapore Eye Research Institute, 11 Third Hospital Avenue, Singapore 168751, Singapore. [9] Centre for Eye Research Australia, Melbourne, Vic, Australia. [10] Ophthalmology, Department of Surgery, University of Melbourne, Melbourne, Vic, Australia. [11] Shenzhen Eye Hospital, Shenzhen University School of Medicine, Shenzhen, China. [12] Center for Individualized Medicine, Department of Molecular Pharmacology & Experimental Therapeutics, Mayo Clinic, Rochester, MN 55905, USA. [13] Department of Physiology, Yong Loo Lin School of Medicine, National University of Singapore, Singapore 117593, Singapore. [14] These authors contributed equally: Pradeep Gautam, Kiyofumi Hamashima. ✉email: Li.Hu@mayo.edu; timothy.blenkinsop@mssm.edu; yhloh@imcb.a-star.edu.sg

The vertebrate eye has undergone substantial changes since humans diverged from earlier species, including earlier mammals and fish. However, the form and function of its compartments that make up the human eye are similar across evolutionary time[1]. Differences that may be present at the cellular level remain unknown, and until recently, the tools to compare species with such resolution were limited. Recently, the development of single-cell RNA sequencing (scRNA-seq) renders transcriptional comparisons feasible.

Each eye tissue plays a crucial role in enabling vision. The anterior segment allows light entry into the eye, controls how much light enters and focuses light to the back of the eye for vision processing. In addition, vision processing occurs in the neural retina (NR), which absorbs the energy in a photon of light, enabling the transformation of energy into electrical force, which then undergoes further secondary order processing before being transmitted to the rest of the brain through the optic nerve. Finally, many tissues of the eye enable efficient nutrient supply, removal of waste, and provide structural support necessary for maintaining normal vision. As a result, disruption of the role in any eye tissue leads to vision problems. Therefore, a study that incorporates as many tissues as possible for comparison between mammalian species is warranted. Although it seems that only one or two ocular tissues are dysfunctional in many of the common eye diseases, the causes for those damages could arise from problems of other ocular compartments. Therefore, a comprehensive study of the optic networks among various tissues is necessary to advance our understanding of the entire eye and its relevant problems.

To date, an eye atlas is still lacking. The human eye is a heterogeneous entity derived from neuroectoderm, neural crest cells and mesenchymal cells with diversity among tissues. For example, within the NR, seven lineages are traditionally described, with many more subtypes. Over the past decade, such heterogeneity has been more and more emphasised. While most previous studies have focused on the whole transcriptome of NR[2–4], the more recent studies have shifted the focus toward understanding the cellular transcriptomic diversity of various ocular tissues. Several scRNA-seq studies describe NR, the retinal pigmented epithelium (RPE) and the choroid[5,6]. Yet, such studies are limited in tissue types and analyses to reveal integration within the ocular entity. Considering this, a more comprehensive transcriptomic description of the ocular components remains incomplete.

Here, we create a multi-species single-cell transcriptomic atlas consisting of the cornea, iris, ciliary body, NR, RPE and choroid. Initial analysis of each tissue type reveal heterogeneous populations, enabling the identification of unique markers and rare cell types, including a transcriptional description of putative stem cell populations of the iris. We look into five aspects of any given cell population and perform an integrative analysis. First, we investigate the transcriptomic similarity of cell populations across species to understand the conservation of cell types. Secondly, we identify transcription factors (TFs) and their target genes active in distinct cell types and compare their conservation among species. Thirdly, we connect ligand–receptor interactions among cell populations to understand the cellular microenvironment and communication pathways activated. As an example, we show the interaction between putative stem cells with different cell populations of the eye. Fourthly, we create a disease map of genes involved in various eye disorders that extend the disease maps beyond retinal cell types. In the fifth place, we report a viral entry map by creating a map of genes that act as entry points for viral invasion into cells. Finally, as a proof-of-concept to show the power of our TF regulon analysis, we investigate one of identified retinal ganglion cell (RGC) TFs, Kruppel-like Factor 7 (KLF7). We perform overexpression (OE) and knockdown (KD) of KLF7 in RGC cells undergoing differentiation and discover that KLF7 drives the maturation of RGC.

## Results
### Identification of cell types in the human and porcine eye.
Cells of different regions of the eye were extracted for scRNA-seq, including iris, cornea, choroid, sclera, retina and RPE (Fig. 1a, Supplementary Fig. 1a). In total, approximately 50,000 cells and 24,075 genes were detected. At least 16 distinct clusters were formed by t-distributed stochastic neighbour embedding (tSNE), an unsupervised graph clustering method (Fig. 1b). Annotation of cell types was based on the literature[7–10] and differential gene profile of each cluster (Fig. 1c and Supplementary Data 1–2). Detailed markers used for cell types are listed in Supplementary Table 1. Characteristic cell type markers were visualised over the tSNE plot to show specific gene expression (Fig. 1d). To add another layer of specificity, we performed a Gene Ontology (GO) analysis of each cluster (Fig. 1e). For photoreceptors (PRs) cells, terms like the sensory perception of the light stimulus were observed, consistent with PRs light-responsive nature. Axon and protein localisation to synapse term was observed in RGCs, accordant with its role in central nervous system connection. Evaluating cell cycle phase genes revealed PRs to have a high proportion of cells in the G2M phase (Supplementary Fig. 1b, e), which might have a role in disc shedding as PRs are known to renew their outer segment by this process[11].

One of the advantages of scRNA-seq analysis is the ability to describe the heterogeneity of a given tissue. Here, we observed the RPE layer is relatively homogenous while other tissues have heterogeneous cell populations (Supplementary Fig. 1d). The number of genes expressed per cell is a good indicator of data quality. Checking the quality of cells throughout the tSNE plot by the number of genes expressed indicated overall good cell quality (Supplementary Fig. 1c). As a quality control, we checked for the expression of gender-specific genes and found that they matched the information from donor data (Supplementary Fig. 1f). To probe the quality of our dataset even further, we checked the S-cones and L/M-cones markers in our cone PRs. Both can be detected (Supplementary Fig. 1g) with L/M-cones comprising a much more significant proportion of the whole cone population (Supplementary Fig. 1h). We also checked the phenotype anomalies that might be caused by the mutation of genes in cell types. Phenotype Ontology of rod PR showed retinal dystrophy, while abnormal iris pigmentation was present for melanocytes (Supplementary Fig. 1i). Such analyses will help genetic studies whose mutations contribute to ocular malfunctions. Samples from six human donors were used to create an ocular atlas (Supplementary Fig. 1j). We performed a comparative transcriptome analysis across species and produced ocular atlases for the pig eye (Supplementary Fig. 1k). In total, based on established criteria for scRNA-seq data, the sequence data used here was of good quality.

### Diversity and conservation of neural retina across species.
Although neural retina on a single cell transcriptomic level has been one of the most well characterised ocular compartments to date (Fig. 2a), new approaches reveal observations not previously understood. In our scRNA-seq data, nine distinct cell types were recognised and sub-classes of bipolar cells (BPs) have been identified (Fig. 2b) based on the canonical markers used for annotation (Supplementary Fig. 2a). First, we checked the similarity of retinal scRNA-seq data with two other published retinal scRNA-seq datasets[5,12]. After that, we found a high correlation between corresponding cell types in the eye (Supplementary Fig. 2c), validating our cell classifications. Next, we reported

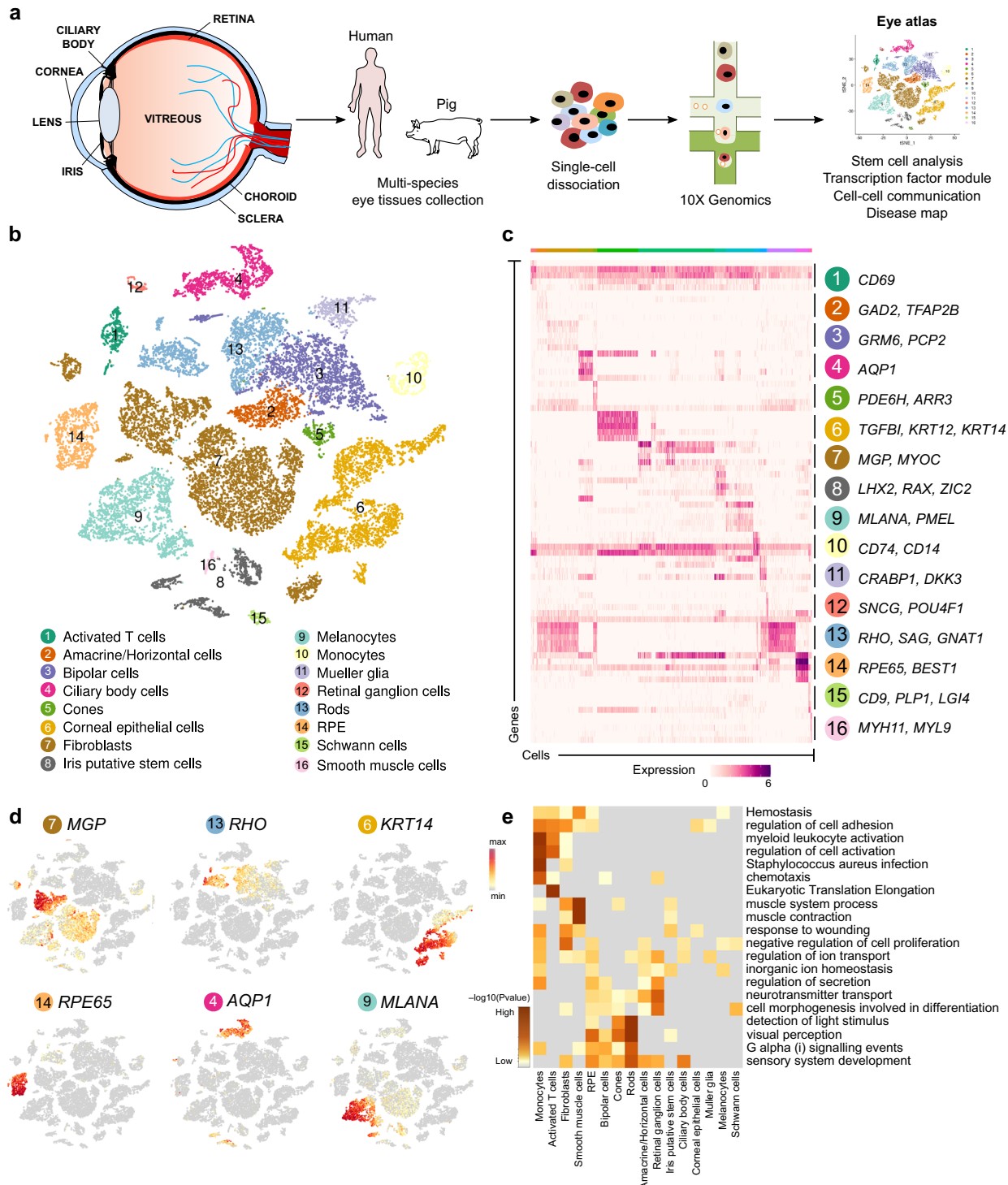

**Fig. 1 Preparation of single-cell transcriptome atlas of the human eye. a** Overview of single-cell RNA-seq libraries prepared from different sources. Postmortem human and pig eyes were enzymatically dissociated, and single cells were isolated. Approximately, 50,000 single cells across the human eye of 6 individuals using droplet-based scRNA-seq platform were profiled. **b** tSNE plot visualisation of human eye cell types coloured by 16 different transcriptionally distinct clusters. **c** Heatmap of differentially expressed genes (DEGs) used to classify cell types for each cluster. The top 5 genes were selected using the one-sided Wilcoxon rank-sum test (*p*-value < 0.01 and |avg_log2FC| > 0.25), and ranked based on their *p*-values within each identified cell type. Scaled expression levels for each cell are colour-coded. **d** tSNE plots showing expression of selected marker genes depicting major classes of cells in the human eye. Scaled expression levels for each cell were colour-coded and overlaid onto the t-SNE plot. **e** GO analysis of DEGs associated with distinct clusters. Metascape calculated the statistical significance of each GO term enrichment (*p*-value) based on the accumulative hypergeometric distribution. The grey colour indicated a lack of significance.

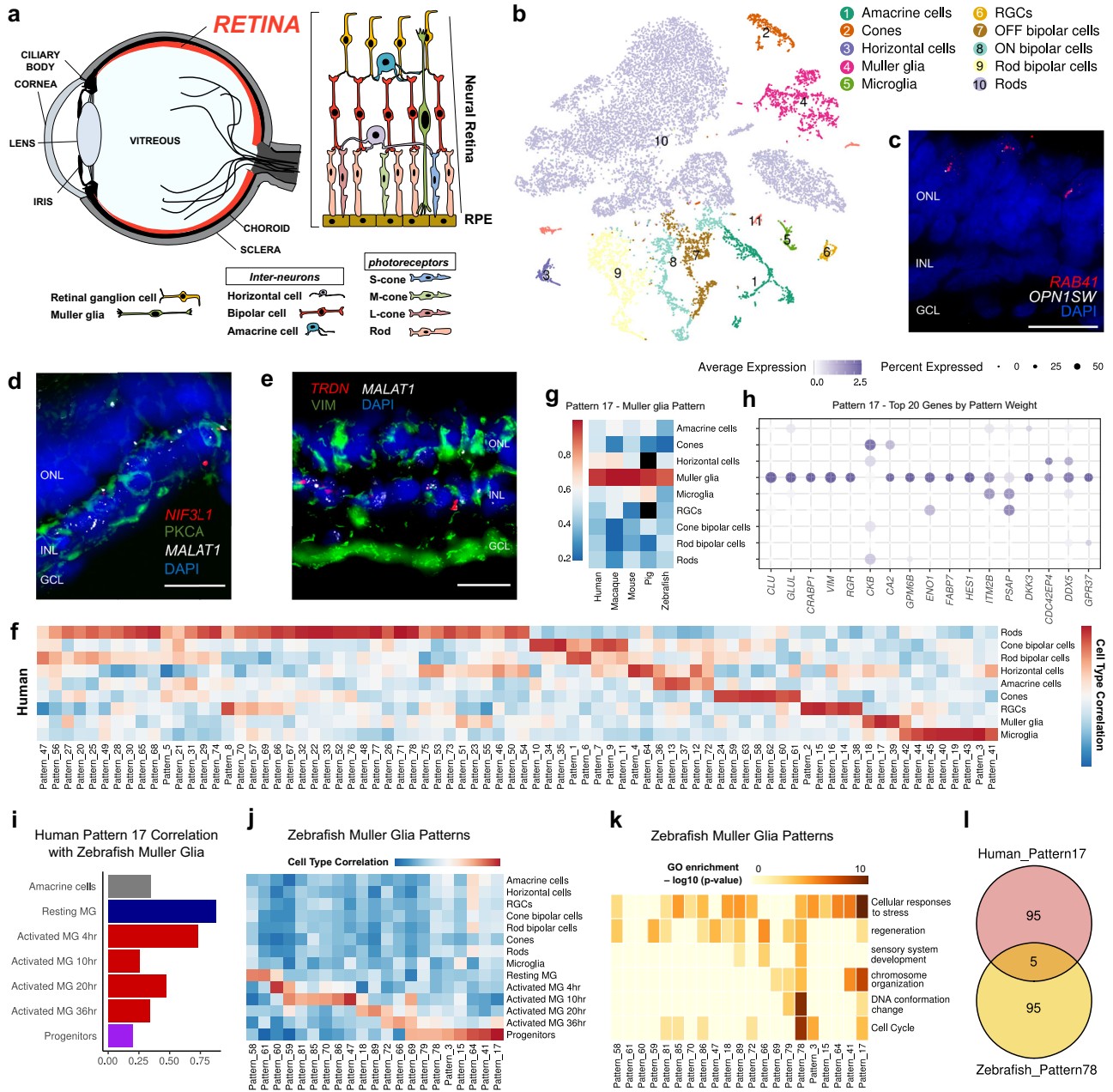

**Fig. 2 Meta-analysis of retinal cells with different donors and species. a** Highlighted region of the eye was selected for single-cell analyses. **b** tSNE plot visualisation of cells obtained from human retina. Ten transcriptionally distinct clusters were observed in the neural retina of the eye. **c** *RAB41* colocalisation with *OPN1SW* using RNA FISH. INL inner nuclear layer, ONL outer nuclear layer, GCL ganglion cell layer. Scale bar = 20 µm. *n* = 2 technical replicates. **d**, **e** Representative RNA FISH images of the novel markers *TRDN* (**e**) and *NIF3L1* (**d**) for different regions of the eye. *MALAT1* was used as an internal control, while *PKCA* was immuno-stained with *NIF3L1* and *TRDN* with *VIM*. Scale bar = 20 µm. *n* = 2 technical replicates. **f** Patterns of gene expression as determined by scCoGAPS algorithm in retinal cell types of the human eye (see Supplementary Information). The correlation of each pattern to human retinal cell types was colour-coded. **g** Pattern 17 showed a high correlation to Muller glial cells across species. **h** Bubble plot showing expression levels of the top 20 genes by gene weight of pattern 17. The size of each circle is proportional to the percentage of cells expressing the gene, and its intensity depicts the average transcript count within expressing cells. **i** Correlation of human pattern 17 with resting Muller glial cells and Muller glial cells activated after injury in zebrafish. **j** The patterns of gene expression in zebrafish Muller glial cells which were activated after injury. **k** Respective GO of the patterns in Fig. 2j. Metascape calculated the statistical significance of each GO term enrichment (*p*-value) based on the accumulative hypergeometric distribution. The grey colour indicated a lack of significance. **l** List of genes that were common between human pattern 17 and zebrafish pattern 78.

markers with unknown roles in retinal cell types (Supplementary Fig. 2b).

As a validation, RNA fluorescence in situ hybridisation (FISH) was performed using human retinal slides on the candidates of novel markers. Chosen markers *NIF3L1* (Fig. 2d) and *TRDN* (Fig. 2e) were confirmed for localisation in human retinal slides.

*TRDN* was shown to colocalise in VIM + cells in Muller glial cells while *NIF3L1* colocalises with PKCA + BPs in RNA FISH experiments.

*MALAT1* localisation could be seen in the inner nuclear layer (INL) layer in both targets as an internal control. *RAB41* was recently shown to localise in cone PRs[13]. As an additional

validation for our RNA FISH results, we offer similar results where *RAB41* localises with *OPN1SW* + cone cells in Fig. 2c. All these novel markers have consistent expression across all the four human donors and may have yet to be explored in terms of functional significance (Supplementary Fig. 2b).

To further explore the NR transcriptome from an evolutionary point of view, we compared our retinal atlas to mouse, primates, and zebrafish retinal atlases[7,14,15] Combinatorial analysis of these datasets enabled the comparison of individual retinal cells across multiple species with the human dataset. We used a single-cell coordinated gene association in pattern sets (scCoGAPS) algorithm to find gene patterns specific to retinal cell types (Fig. 2f). The patterns specific to human retinal cell types were projected into the retina of other species (Supplementary Fig. 2d).

Some pattern-cell type combinations are conserved across species, for instance, pattern 13 for amacrine cells (ACs) (Supplementary Fig. 2e), pattern 24 for cone cells (Supplementary Fig. 2f), pattern 71 for rod cells (Supplementary Fig. 2j), and pattern 6 for rod bipolar (Supplementary Fig. 2k) cells. However, another pattern-cell type like pattern 34 for cone BPs was only found in mammalian species (Supplementary Fig. 2g). Some other patterns of interest were Pattern 4 for horizontal cells (Supplementary Fig. 2h) and Pattern 2 for RGC cells (Supplementary Fig. 2i). All patterns specific to cell types are also listed in Supplementary Data 3. Muller glial cells are of particular interest among all retinal cell types due to their known species difference. The excellent regeneration ability to various neurons in zebrafish marks its significant difference from the mammalian species[16]. To understand this, we found patterns specific to Muller glial cells across the species. And among such patterns was Pattern 17 (Fig. 2g), and the genes that constitute Pattern 17 were shown in Fig. 2h. Some of the genes which were included in Pattern 17 were removed (*RHO*, *ACTB*, and *GAPDH*) as they were considered artefacts of the scGoPAS algorithm. We checked the correlation of Pattern 17 to zebrafish Muller glial cells at different stages of activation after injury. We found out that Pattern 17 matches the most with resting Muller glial cells of zebrafish. The similarity decreases as they transit to form progenitors from resting Muller glial cells (Fig. 2i, j). The gene expression patterns that appear in Muller glial cells activation after injury show GO terms like "regeneration", "sensory system development" and "cell cycle" (Fig. 2k). One similarity of such patterns between zebrafish and humans (like pattern 78 in zebrafish and pattern 17 in humans) was none, as they have very few genes in common (Fig. 2l). Module 17, which was conserved across species, does not include gene patterns related to regeneration.

**Characterisation of non-retinal ocular structures**. Non-retinal ocular structures help support the structure of the eye and manage light hitting the retina. The sclera provides protection and structure to the eye, while the choriocapillaris (Choroid) is the vascular bed underlying the Bruch's membrane providing nutritional support for the retina. The single-cell map of the choroidal and sclera layer (Fig. 3a, Supplementary Fig. 3a) shows that cells that populate the sclera are fibrotic tissue. The choroidal layer of the eye consisted of endothelial cells and fibroblast cells. Since it is difficult to separate these two tissues physically and share cell types in common, we analysed them together (Supplementary Fig. 3a). We observed cell types that typically circulate in the choroidal vessels, including activated T cells and monocytes (Fig. 3a). The canonical markers used for annotation of cells and novel markers for those cell types provide additional resources for further study into diseases of the sclera and choroid (Supplementary Fig. 3d).

The cornea is the outermost transparent layer of the eye, whose primary function is transparency and to act as a barrier. The adult cornea has three layers: an outer epithelium (ectoderm), a middle layer containing a collagen-rich stromal region composed of fibroblast cells, and an endothelial cells' inner layer. Corneal fibroblasts are originated from neural crest cell[16]. In our dataset, the cornea is populated mainly by two epithelial cells (Fig. 3b, Supplementary Fig. 3b). One of them have high expression of the *TGFBI* gene than the rest of the cells, and another has high expression of the *ELF3* gene (Fig. 3b, Supplementary Fig. 3e). *ANXA1* is a marker of cells undergoing inflammation[17] is expressed highly in TGFBI-high epithelial cells (Supplementary Fig. 3e), while *ELF3* is expressed in differentiating corneal epithelial cells[18]. Corneal wound healing involves inflammation, proliferation and differentiation processes[19] and expression of *TGFBI* and *ELF3* can distinguish which stage any given corneal cell is present. Such processes might be visualised by the RNA velocity analyses of corneal cells (Supplementary Fig. 3g). Other corneal cell types include fibroblasts, melanocytes, monocytes, cytotoxic T cells, and conjunctival cells. The canonical markers used for annotation of these cell types and the novel markers associated with them are in Supplementary Fig. 3e.

The iris functions to regulate the amount of light reaching the retina. The anterior portion of the optic cup of the eye during development gives rise to iris epithelium and ciliary body epithelium[8]. The stromal region of the iris is generated from neural crest cell migration[20]. However, the smooth muscles of the iris are developed from optic cup neuroectoderm, and ciliary muscles responsible for changing the shape of the lens are made from surrounding mesenchymal cells[9]. As a result, iris tissue has heterogeneous cells derived from different developmental origins. We found that most cells were fibroblasts from our scRNA-seq data of the iris/ciliary body (Fig. 3c, Supplementary Fig. 3c). They were subdivided into fibroblasts, MEG3-high fibroblasts, MGP-high fibroblasts, WIF1-high fibroblasts and Ribosomal genes high fibroblasts. Fibroblasts highly expressing *MEG3* is proliferating in glaucoma tenon fibroblast[10].

Similarly, fibroblasts having high expression of *WIF* have been shown to initiate melanogenesis in normal human melanocytes[21]. Iris stroma is also populated by Schwann cells that help myelination of axons of neuronal cell types that populate iris[22]. They expressed *CD9* marker, which is implicated in the signalling of Schwann cells with axon[23].

Ciliary body cells (CBCs) help produce aqueous humour in the anterior chamber of the eye[24]. CBCs were subdivided into CBCs, COL9A1- high CBCs, CRYAA-high CBCs and pigmented CBCs (Fig. 3c). Smooth muscle cells (SMCs) that populate sphincter pupillae and dilator pupillae muscles were detected in the data (Fig. 3c). They had high *MYH11* and *MYL9* (Supplementary Fig. 3f) expression, which are the canonical markers for SMCs[25]. A pigmented layer of epithelial cells also populates the iris. They expressed canonical markers involved in pigmentation like *MLANA* and *PMEL* (Supplementary Fig. 3f).

We tried to visualise the cell-cell interaction among all cell types of the eye. As expected, we found more interactions with cells located physically together (Supplementary Data 4). However, monocytes were also seen to have more interactions with the cell types in the retina. We found the most significant interacting pairs (Fig. 3d) and clustered them using the tSNE plot (Fig. 3e). The tSNE plot helps to give an idea about the similarity of cell–cell interactions among cell types. We aimed to provide a global picture of cell–cell interactions across cell types, and Fig. 3f showed these kinds of interactions.

Also, we checked cell types having the most interactions in the whole eye. The most interactions were with Muller glial cells across species (Supplementary Fig. 3h). However, such analysis is

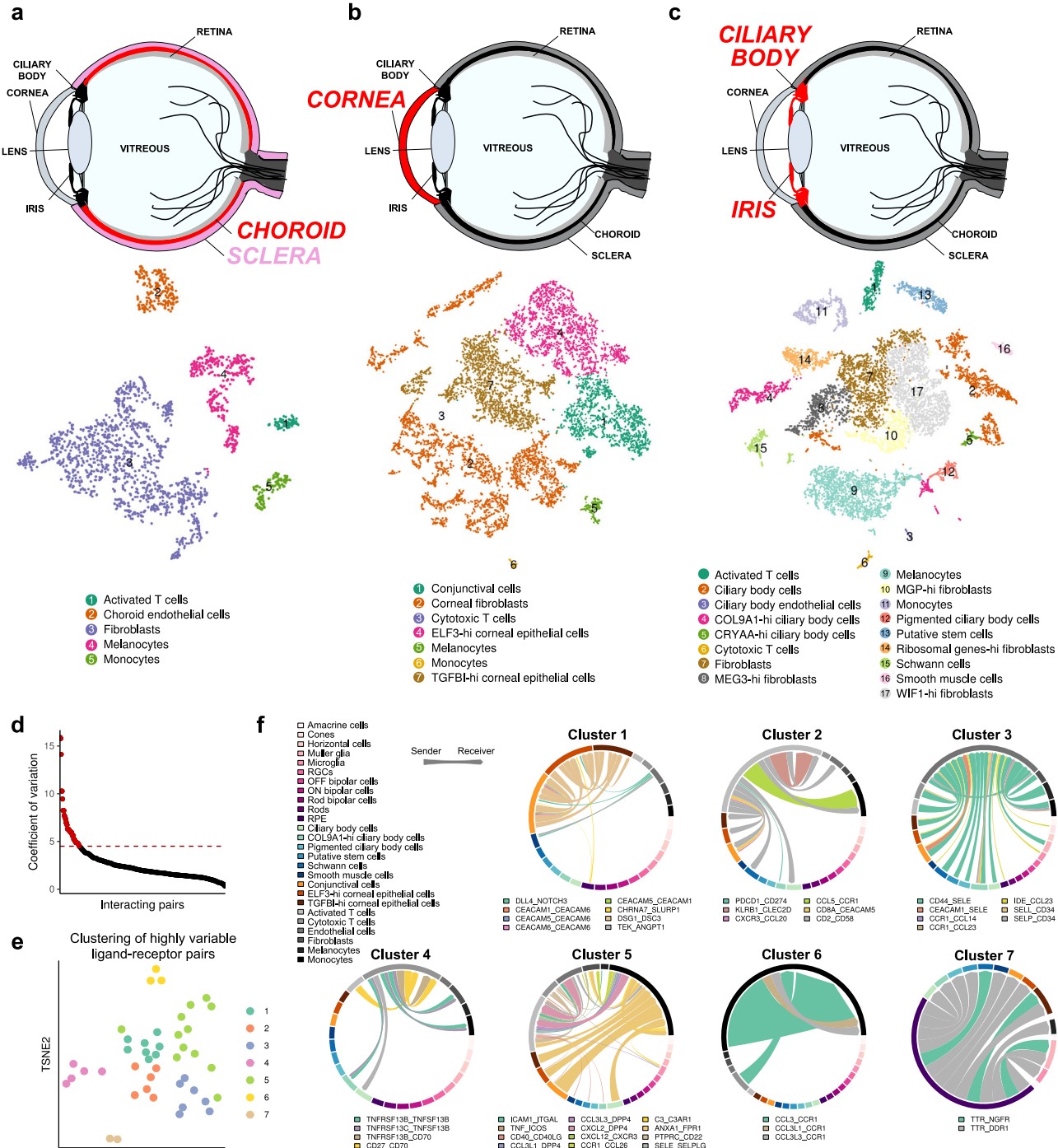

**Fig. 3 Single-cell transcriptome atlas from non-retina cells in the human eye. a** tSNE plot visualisation of cells obtained from scleral and choroid layers of the eye. Five transcriptionally distinct clusters were observed. **b** tSNE plot visualisation of cells obtained from the cornea of the eye. Seven transcriptionally distinct clusters were observed. **c** tSNE plot visualisation of cells obtained from iris pigmented epithelium, iris muscle, and stromal region of iris. Seventeen transcriptionally distinct clusters were observed in the iris region of the eye. **d** Selecting highly variable interacting pairs that exhibited high cell type-to-cell type variation in the dataset. **e** Hierarchical clustering of similar cell–cell signalling probability scores and visualised on a tSNE plot. **f** Global cell–cell interaction map across cell type of the eye. Edge weights represent the probability of signalling between cell clusters.

limited by the genes selected for the study, which were paralogs of human genes across species. As a result, we could see lesser interactions in other species compared to humans.

**Putative stem cell populations of the iris.** Among various cell types in the iris/ciliary body, we could detect a distinct population of cells that express markers of stem cells. Even though these cells express markers of retinal progenitor cells or retinal stem cells,

such cells need to give rise to retinal neurons in vivo to label them as retinal stem cells. The absence of in vivo experiments limits our work. Previous studies have mentioned that Muller glial cells share 60% of the transcriptome with retinal progenitors in mice[26]. However, transcriptomic similarity or the presence of specific genes markers do not guarantee if a cell is a retinal stem cell or not. Therefore, such cells remain as putative stem cells for now.

Besides putative stem cells, two subpopulations of CBCs, COL9A1 high CBCs and pigmented CBCs showed higher similarity to such putative stem cells. All three cell types expressed of *PAX6* and *SIX3*, eye field TFs (EFTF). Putative stem cells have an expression of *OTX2*, which is another EFTFs (Supplementary Fig. 3f). COL9A1 high CBCs expressed *CPAMD8* (Supplementary Fig. 3f), which plays a role in periocular mesenchyme development[27]. We also found the presence of both cells in the iris of pigs (Fig. 4c). Checking for stem cell potency across cell types showed the identified stem cell populations score higher on a stem-cell potency index (Fig. 4a, b), providing evidence that such cell populations may exhibit stem cell properties. RNA velocity is a high dimensional vector that predicts the fate of cell populations in scRNA-seq data[28]. Here, RNA velocity analysis showed all three putative stem cells cluster together in Uniform Manifold Approximation and Projection (UMAP) plots (Supplementary Fig. 4a). It is suggestive of transcriptional similarity hinting that pigmented CBCs and COL9A1-high CBCs might originate from putative stem cells. We tried to understand the gene patterns that make up the putative stem cells, pigmented CBCs and COL9A1-high CBCs. We found out that patterns 13, 37, 48, 54, 55, 56, 57, 86 and 87 was shared among three cell types (Fig. 4d). When these gene patterns were projected into pig iris cell types (Supplementary Fig. 4b), we could see that specific gene patterns like patterns 13, 87, 56 and 86 were conserved across the species (Fig. 4d, e). Upon further examining the type of biological processes these gene patterns constitute, we investigated the GO terms that were enriched in such gene patterns. GO terms like "Neural Crest Differentiation", Negative regulation of cell differentiation", and "embryonic morphogenesis" were enriched (Fig. 4f). One of the patterns specific to COL9A1 high CBCs, Pattern 56, was probed for the genes it comprised (Fig. 4g). Previous research has pointed that there are multipotent cells derived from neural crest in adult mouse iris[29]. So, this suggests that COL9A1 high CBCs might be some sort of multipotent stem cells.

**Ligand–receptor interactions in putative ocular stem cells**. We focused our analysis on signalling pathways involved in stem cell maintenance to identify which primary molecules are at play in the eye. We focused on three signalling pathways, i.e., Fibroblast growth factor (FGF), WNT and Midkine (MDK) signalling pathways (Fig. 4i–k). Midkine, one of the ligands specific to retinal progenitor cells in zebrafish, showed high expression in putative stem cells[30]. It was also shown to mediate glial activity, neuronal survival and the reprogramming of Muller glia into proliferating Muller glial proliferating cells in chicks[31]. *MDK* expression was high in putative stem cells, and its receptors were shown to be present in both neuronal and non-neuronal cell types of the eye (Fig. 4k). Thus, the expression of *MDK* added to the evidence of putative stem cells being present in iris tissue. When we focussed on the WNT signalling pathway, gene expression of *ROR1*, *ROR2* and *FZD1* receptors were found to be distributed among putative stem cells, pigmented CBCs and COL9A1 high CBCs (Fig. 4h). MAGIC co-expression analysis based on MAGIC imputation revealed WNT receptors *ROR1*, *ROR2*, and *FZD1* were specific to pigmented CBCs, putative stem cells and COL9A1-high CBCs, respectively. These cells were also uniquely co-expressed with *FGFR1* (Supplementary Fig. 4e). *FGFR1* expression was also conserved across similar cell types in pigs (Supplementary Fig. 4c, d).

**Creation of disease map and viral-entry map for the human eye**. We wanted to provide a resource to understand the disease map and viral-entry map of cell types in the human eye. For that,

we obtained a list of genes that were affected in ocular malformations and checked the gene expression of those genes across the whole eye. We focused on genes that cause colour blindness, corneal disorders, eye cancer, eye movement disorders, macular degeneration, optic nerve disorders, retinal disorders, vision impairment and blindness (Fig. 5a). For example, *GSN* mutation causes reduced corneal sensitivity in the later stage of life. Mutations in *GSN* causes deposition of amyloid in a different part of the eye[32] However, expression of *GSN* throughout the cell types of the eye shows that it is expressed in CBCs, SMCs and fibroblasts in the eye (Fig. 5b). It gave an idea of how mutations of *GSN* could cause corneal dystrophy.

We also checked the expression of such genes across species. We showed that genes for cone/rod dystrophy, retinitis pigmentosa and stationary night blindness are conserved across species (Supplementary Fig. 5a). We looked into the viral entry map of the human eye. As the world's interest in COVID-19 increases, we wanted to provide a map of cell surface proteins that can act as viral entry receptors. We showed that *ACE2* and *TMPRSS2*, the primary cell surface proteins responsible for entry into human[33], are expressed in the cornea's conjunctival cells (Supplementary Fig. 5b, c). Other receptors of interest like *BSG*, *CTSB* and *CTSL* were also shown. As a resource, we provided information for the cell surface proteins that serve as an entry point for other classes of viruses (Supplementary Fig. 5b).

**Unique transcriptional regulons active in the human eye**. Since TFs orchestrate gene expression across the genome, cell identity and function can be partially described by the expression of its TFs. The TFs expressed may provide insight into the machinery that maintains their stemness with a focus on the putative stem cells. We made a pipeline to understand transcriptional regulons active in different retinal and putative stem cells. We combined the Regulon activity score computed from SCENIC and gene imputation scores calculated from MAGIC to create a pairwise correlation of TFs active in cell types (Supplementary Fig. 6a). The TFs which were enriched in cell types are also listed in Supplementary Data 5. We found 9 modules specific to cell types in the retina (Fig. 6a, Supplementary Fig. 6b). Module M1 was specific to RGCs, and module M5 was specific to PR cells and presented in Fig. 6b, c, which included the responsible TFs. The specificity of such modules in cell types could be seen in tSNE plots (Fig. 6d). With the help of a correlation matrix among TFs, we plotted the interactions of TFs with each other in modules (Fig. 6e).

Analysis of the correlation map showed that TFs active in Schwann cells and Muller glial cells were highly correlated as both are glial cell types. The pigmented CBC types and retinal pigmented epithelial cells were also highly correlated as both are pigmented cells. Besides checking the transcriptional modules of retinal cells, we evaluated the transcriptional modules of non-retinal cell types (Supplementary Fig. 6c–e) of the eye. The TFs that populate melanocytes of Iris included *PAX3*, *MITF* and *SOX10* (Supplementary Fig. 6e). These factors have been shown to be important in the *trans*-differentiation of fibroblasts into melanocyte[34]. Thus, such TF modules might also provide a resource for *trans*-differentiation of cell types.

**Conservation of TF modules across species**. We checked for the genes that were involved in the formation of TF modules. To do so, we conducted a GO analysis of the TFs, and their target genes confirmed the GO terms are specific to each cell type (Fig. 7a, Supplementary Data 6). To gain additional confidence in the presence of TF module specific to cell types, we verified *SREBP2* and *KLF7* specificity in RGCs and *PBX1* in ACs and RGCs using

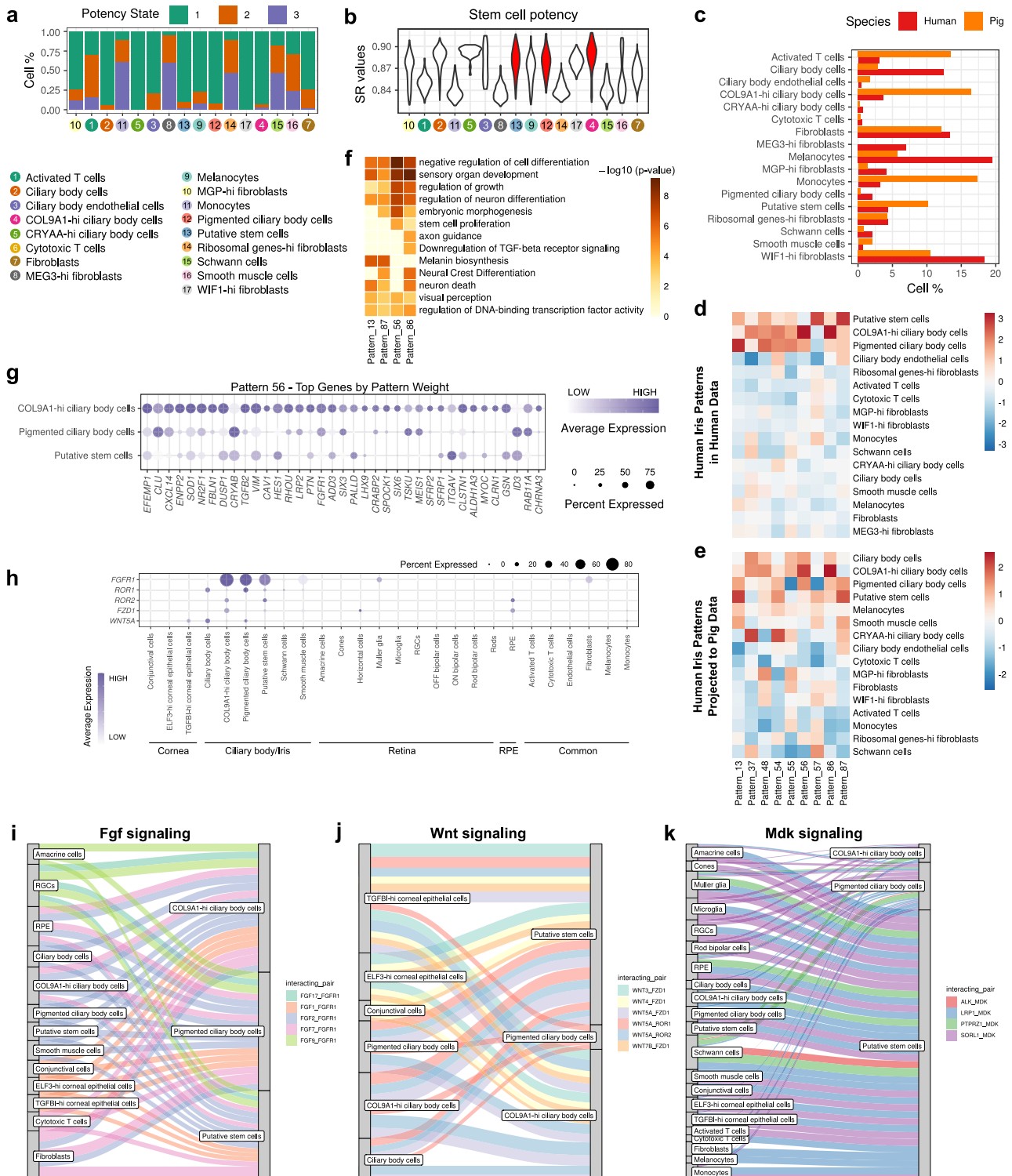

**Fig. 4 Characterization of putative stem cells. a, b** Stem cell potency of cell types in the iris region of the eye. The stem cell potency scores (SR values) and potency states were inferred using SCENT (see Supplementary Information). **c** Similar proportion of cell types in iris could be detected across a pig and human samples. **d, e** Patterns of gene expression as determined by scCoGAPS algorithm in iris cell types of the human eye (**d**) and projection of those patterns into pig iris cell types (**e**). Nine patterns highly correlated with either putative stem cells, COL9A1-high ciliary body cells or pigmented ciliary body cells were selected. **f** GO enrichment terms for patterns specific to COL9A1 high ciliary body cells, pigmented ciliary body cells and putative stem cells. Metascape calculated the statistical significance of each GO term enrichment (p-value) based on the accumulative hypergeometric distribution. The grey colour indicated a lack of significance. **g** Genes that make up the patterns specific to COL9A1 high ciliary body cells. **h** Expression of receptors specific to putative stem cells, pigmented ciliary body cells and COL9A1-hi ciliary body cells. **i–k** Interaction map between FGFs (**i**), WNTs (**j**), and MDK (**k**) secreted by several cell types with stem cells in the eye, respectively. Edge weights represent the probability of signalling between cell clusters.

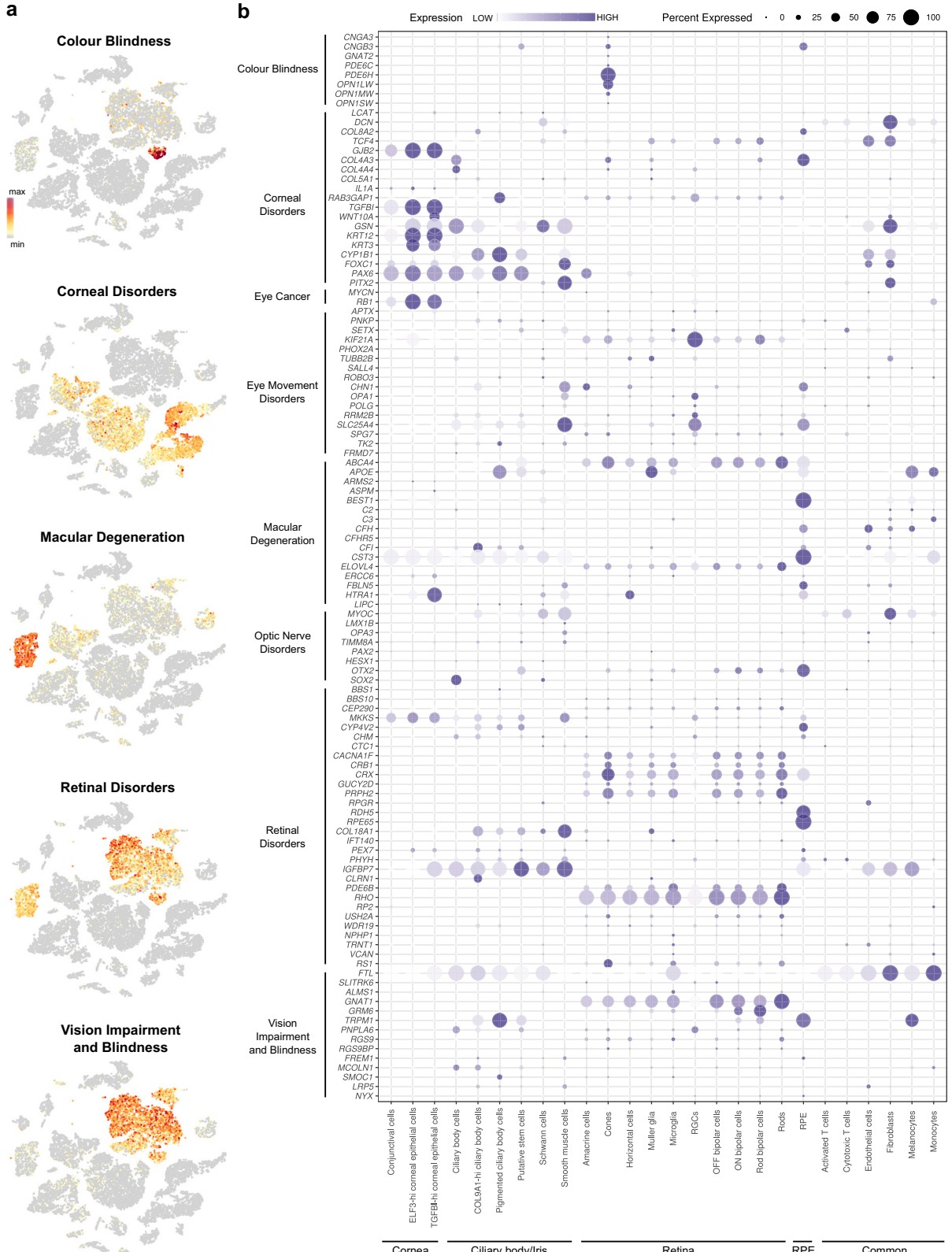

**Fig. 5 Creating a disease map for the human eye. a** Human eye disorder-associated gene-set scores visualised on a tSNE plot. Scaled scores for each cell were colour-coded. **b** Bubble plot showing expression of genes involved in different eye disorders across cell types of eye.

RNA FISH. *SREBP2* and *KLF7* are two TFs present in Module 1 specific to RGCs (Fig. 6e).

*SREBP2* is TF involved in cholesterol biosynthesis[35] and *KLF7* involved in the axon regeneration response after optical nerve injury in the eye[36]. We detected the localisation of *SREBP2* and *KLF7* in RGC cells (Fig. 7b, Supplementary Fig. 7a). *PBX1* localisation to the INL layer where amacrine cells were located could also be observed. *RLBP1* was used as a control where it

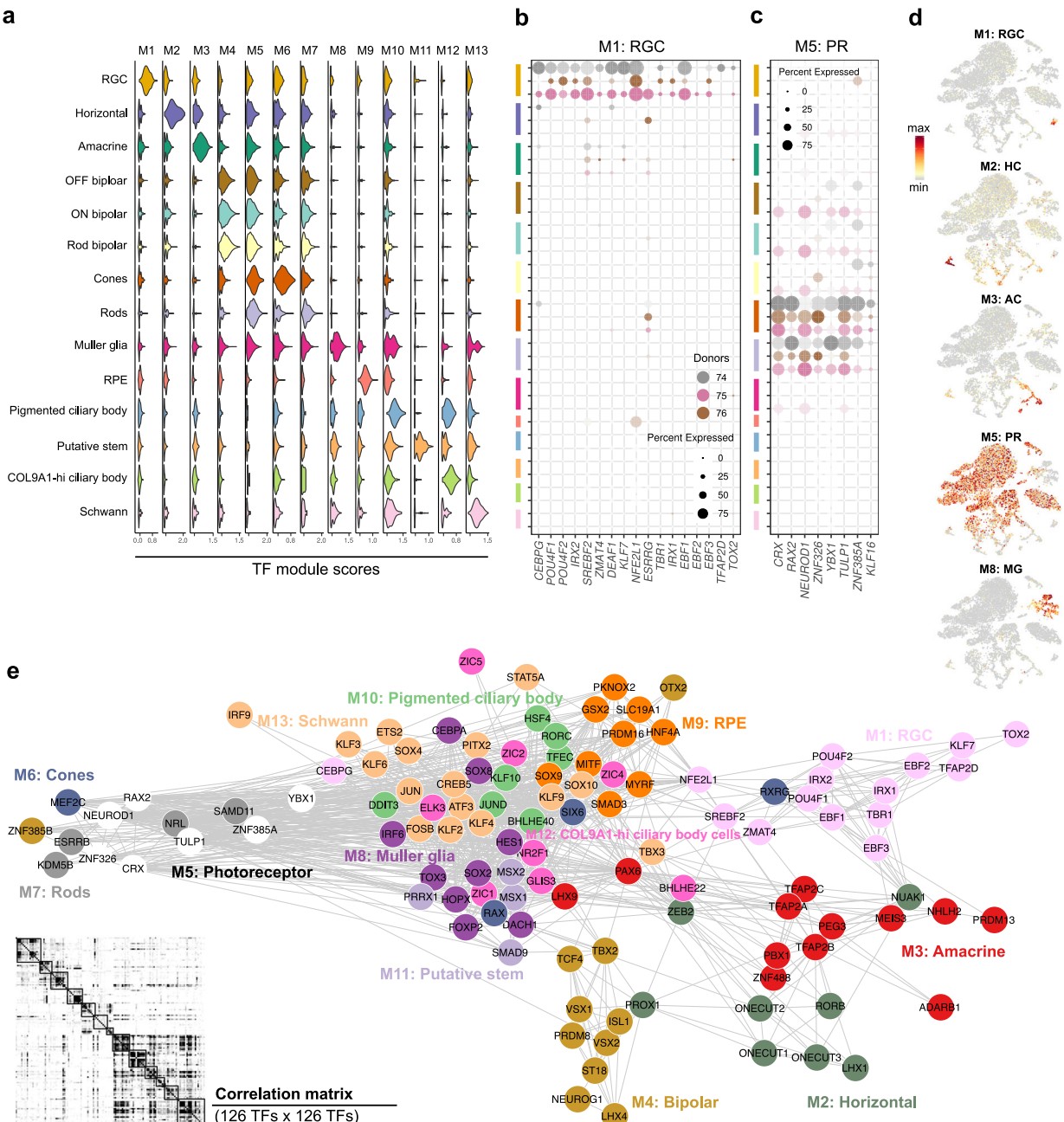

**Fig. 6 Reconstruction of transcriptional regulons that are active in different neural/glial cell types of the human eye. a** Violin plots showing activities of the identified transcription factor modules scores in each cell type. Rows correspond to cell types obtained from different donors, while columns correspond to TF modules specific to a particular cell type. **b**, **c** The representative bubble plot for M1 module (**c**) and M5 module (**d**) were specific for RGCs and PRs, respectively. Rows correspond to cell types from different donors, and columns correspond to the TFs that are part of each module. **d** Regulon activity of selected modules visualised on tSNE plot. Scaled scores for each cell were colour-coded. **e** TF network in the different neural/glial cell types of the human eye. TFs that belong to the same module (shown in the same colour) were clustered together. The correlation matrix of the TFs involved in the formation of 13 different TF modules in the human eye is shown in the corner.

localised to the RPE layer (Supplementary Fig. 7a). Besides RNA FISH, we also conducted immunofluorescence in the primate retina for the TF specificity. We checked whether genes involved in TF modules are sufficient to separate cell types across species and PCA analysis and demonstrated clear separations between cell types based on the unique TFs (Supplementary Fig. 7b, c) compared to randomly selected some TFs (Supplementary Fig. 7d, e). Such conservation of TF modules could be shown across species (Supplementary Fig. 7f) using pairwise correlation. The TFs that were involved in the formation of modules were

checked for their conservation. For example, pairwise correlation plots show that *LHX9*, *TFAP2C* were conserved across species for amacrine cells.

Similarly, *KLF7* was shown to be preserved in RGCs across humans, mice, macaque and zebrafish (Supplementary Fig. 7f). Such analysis helped us to understand the TF differences in cell types across the species. In Klf7-null mice, a small portion of RGCs showed aberrant projections while exiting retina[37]. Klf7 and Pou4f1, another member of RGC module TF, co-operate to control TrkA expression in sensory neurons[38]. Since *KLF7* is also

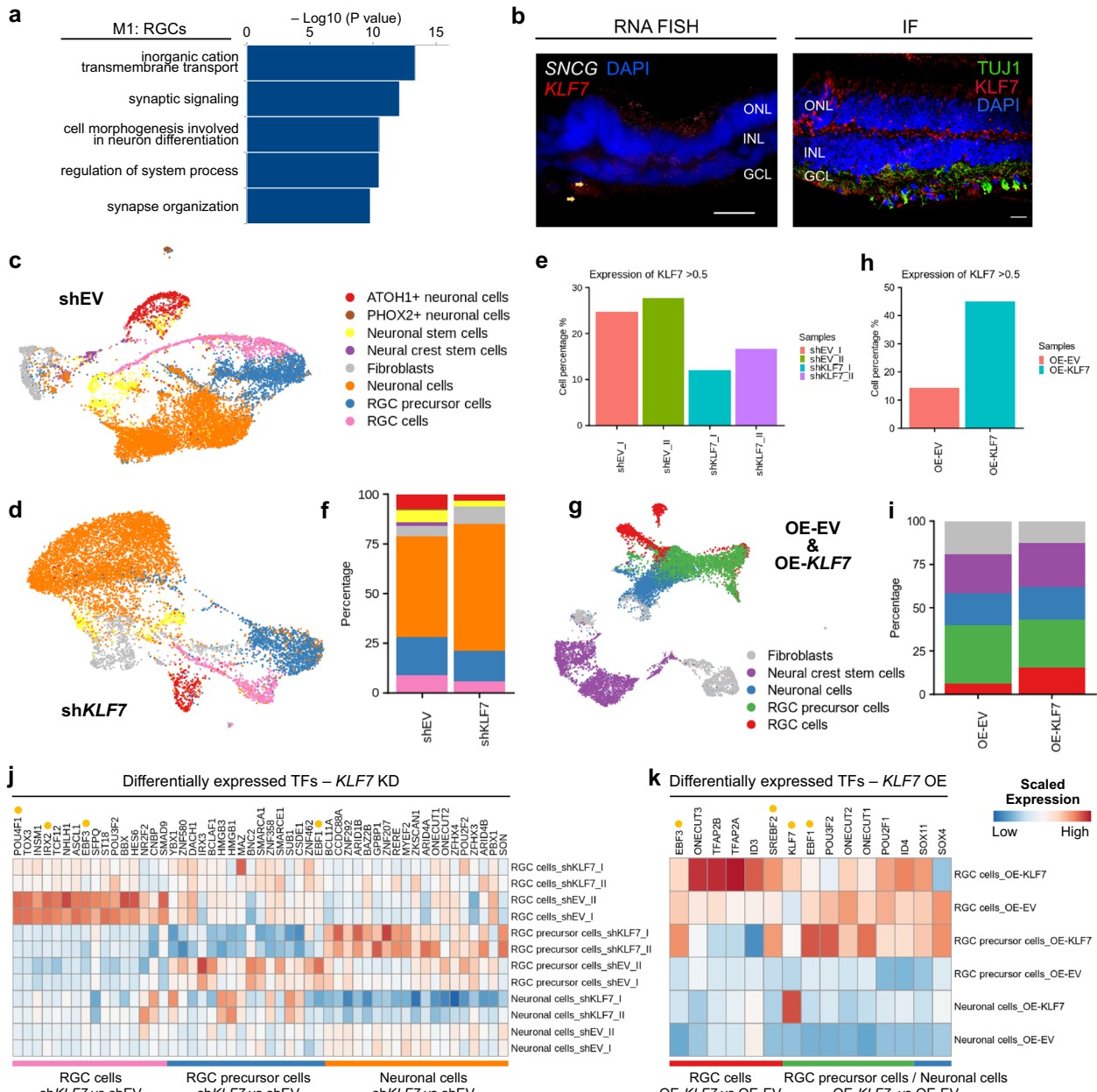

**Fig. 7 *KLF7* acts as a driver for RGC maturation. a** GO analysis of TFs and their targets specific to M1 modules. Metascape calculated the statistical significance of each GO term enrichment (*p*-value) based on the accumulative hypergeometric distribution. **b** Representative RNA FISH images of the RGC-specific TFs *KLF7*. n = 2 technical replicates. Immunofluorescence of the *KLF7* with *TUJ1* was also shown in the non-human primate retina. n = 2 technical replicates. Scale bar = 20 μm. **c** UMAP plot visualisation of cells obtained from differentiating RGC cells transfected with empty vector as control. Eight transcriptionally distinct clusters could be observed. **d** UMAP plot visualisation of cells obtained from differentiating RGC cells which were transfected with shRNAs for *KLF7*. Seven transcriptionally distinct clusters could be observed. **e** Knockdown levels in sh*KLF7* after transfection. **f** RGC cell proportion decreased in sh*KLF7* libraries compared to shEV libraries. **g** Combined UMAP plot visualisation of cells obtained from differentiating RGC cells transfected with empty vector as control and *KLF7* open reading frame. Five transcriptionally distinct clusters could be observed. **h** *KLF7* expression levels after transfection in overexpression experiments. **i** RGC cell proportion increased in *KLF7* OE libraries compared to EV-OE libraries. **j**, **k** Differentially expressed genes (DEGs) across different cell types in sh*KLF7* and shEV (**j**) or *KLF7*-OE and EV-OE libraries (**k**). DEGs were selected using the one-sided Wilcoxon rank-sum test (*p*-value < 0.01 and |avg_log2FC| > 0.25). The genes that have a dot behind them were the M1 module of TFs specific to RGCs.

expressed in foetal RGC during RGC development in humans[39], conserved across species and has a role in axon regeneration response after injury, we wanted to focus on the part of *KLF7* in RGC differentiation and maturation.

***KLF7* acts as a driver for RGC maturation**. We used a protocol[40] to differentiate H9 Human embryonic stem cells into RGCs. We

performed KD experiments in cells that were driven to RGC lineage (Fig. 7c, d, Supplementary Fig. 7g, h). Successful KD was achieved by shRNA transfection (Fig. 7e). Since cells were still in the process of differentiation, mature RGC markers were lowly expressed. However, markers like *POU4F1* and *EBF3*, which act as markers for maturing RGC[41,42], were present in cells (Supplementary Fig. 7j). Cells expressing *ONECUT1*, *ONECUT2* and

*EBF3* were designated as RGC precursor cells because *ONECUT* TFs are expressed in developing RGCs[43].

As a result of the KD of *KLF7*, we detected drastic differences in the proportion of cells that are destined to be RGCs (Fig. 7f). Also, genes like *POU4F1*, *IRX2* and *EBF3*, which comprised the TF modules of RGC (M1 module), were downregulated after KD. Moreover, in RGC precursor cells, decreased expression of *EBF1*, an M1 module TF, was observed (Fig. 7j). To gain further insights, we overexpressed *KLF7* during the early differentiation window (Fig. 7g, Supplementary Fig. 7i), which aimed to verify whether the acceleration of RGC differentiation can be achieved by *KLF7* alone. OE was confirmed (Fig. 7h) after manual annotation with the same criteria used for KD libraries (Supplementary Fig. 7k). Expectedly, the early differentiation window did not yield many mature RGCs. However, the proportion of RGC-like cells was increased after *KLF7* OE (Fig. 7i). Moreover, *KLF7* OE also increased the expression of RGC module TFs, like *EBF3*, *SREBF2* and *EBF1* in RGC precursor cells (Fig. 7k). Those changes hint that *KLF7* might have a role in biasing cell fate toward RGCs during retinogenesis.

## Discussion

Single-cell studies have revolutionised the field of understanding human organs. There have been many cell atlases of different tissues[12,44,45]. They have also been used to understand the dynamic processes of differentiation and reprogramming[46,47]. The data presented here described a detailed reference transcriptome of the eye's cell types, including the cornea, iris, ciliary body, NR, RPE and choroid cell types. Even though transcriptomic atlas for retinal cell types was previously described, we sought to produce an atlas that included non-retinal eye cell populations. We compared our retinal cell types with single-cell studies done earlier to confirm that dissociation and sample processing methods did not cause transcriptomic changes in cell types.

Firstly, comparing the cell types with earlier studies[5,12] showed a strong correlation between the cell types with the published datasets. The retina is a widely studied eye tissue in different species by previous scRNA-seq studies by Macosko et al.[7], Peng et al.[48] and Hoang et al.[15]. Therefore, we compared the transcriptomic differences with human cell types with the help of data from the scRNA-seq of the retina in different species.

We used the scCoGAPS algorithm to find gene patterns specific to the cell type of the eye. The patterns that were specific to human cell types were projected into the retina of other species. Some patterns were specific to cell types which were conserved across species. Looking at some of these patterns, we understood how Muller glial cells in mammals differ from zebrafish. We also checked for TF conservation across species using pairwise comparison. Some of the TFs specific to modules were conserved across all species.

The cells obtained from Iris/ciliary body tissue contained cells from the pigmented epithelial region, iris muscles, CBCs, and the cells from the ciliary margin zone (CMZ). Here, we propose the description of putative stem cell populations in the iris. The presence of adult stem cells in CMZ of the eye has been a subject of debate. Previous studies have shown that the ciliary epithelium in mouse eyes can give rise to clonogenic spheres, giving pan-neuronal marker expression in culture. However, they prove that those CE derived spheres are not retinal stem or progenitor cells[49,50].

Studies in zebrafish CMZ have shown that Wnt2b controls retinal cell differentiation[51]. Furthermore, Wnt2b has also been shown to inhibit neuronal differentiation and induce an immature progenitor state when ectopically expressed in the retina in ovo[52]. Similarly, some Msx1/Msx2 positive cells in the CMZ zone possess retinal progenitor-like properties, giving rise to both neural retina and non-neural ciliary epithelial cell types[53].

Here, we found a distinct population of cells that may express transcripts associated with stem cell properties. COL9A1 high CBCs, pigmented CBCs and putative stem cells have increased expression of *PAX6* and *SIX3*, which are EFTF. Putative stem cell also has high expression of *OTX2*, which is another EFTF. Expression of *PAX6* and *SIX3* is also high in all four CBCs. The stem cell potency scores of all four CBCs are high in the human eye, showing that these cell types could give rise to spheres with colonogenic potential in culture media. However, COL9A1 high CBCs and pigmented CBCs cluster most closely with putative stem cells rather than other CBCs. COL9A1 high CBCs express *CPAMD8*, which was present in the ventral iris which was incompetent in regeneration after lentectomy[27]. Some studies have also suggested *CPAMD8* might have a role in the crosstalk between the optic cup peripheral neuroretina and periocular mesenchyme during ocular development[54]. But the expression of markers specific to fibroblasts like *OPTC* and *COL9A1* in such cells shows that this cell type could be the progenitor of mesenchymal cells in the human eye. Expression of TFs like *NR2F1*, *ZIC1*, *ZIC2*, *ZIC4*, *ZIC5*, *ELK3* and *GLIS3* (Supplementary Fig. 6b) suggests neural crest origin[55–61].

RNA velocity analysis was used to predict the fate of retinal stem cells in retinal cells. The RNA velocity of all ocular cell types shows that COL9A1 high CBCs and pigmented CBCs could be generated from putative stem cell type. The GO analysis of patterns specific to these cells also show terms like "neural crest differentiation". Therefore, COL9A1 high cells may be a neural crest-derived multipotent glial/neuronal stem cell population.

Preliminary analysis revealed several receptors and ligands involved in the WNT and FGF signalling pathway specific to the putative adult stem cells. The *FGFR1* receptor populates them. Studies that focussed on ciliary stem cell-like cells by using organoid models showed that inhibition of FGFR receptors pushed cells toward the RPE lineage. Similarly, removing this inhibitor reversed the fate of RPE lineage to convert into neuroretina-like fate[62]. However, due to the lack of in vivo experiments, such cells are annotated as putative stem cells.

We found 13 different TF modules specific for neuronal/glial cell types in the eye. TFs within the same modules have a high correlation in terms of gene expression. This co-expression network revealed that TFs shared between cell types have more connections with each other. Using this method, we provide a method to extract the genes that define cell identity. Comparing the genes involved in modules unique to cell populations across species helped illuminate which modules are conserved across species. Interestingly, the TFs that make up a module may not necessarily be conserved.

We picked the *KLF7* TF for further analysis in our studies because it has been shown to have a role in RGC protection after injury by regenerating axons[36]. Similarly, it was also expressed in both foetal RGC cells and adult RGC cells. It was also conserved across several species. KD of *KLF7* delayed differentiation of RGC cells as RGC maturation markers went down. Several TFs that make the M1 module (*POU4F1*, *IRX2*, *EBF3* and *EBF1*) specific to adult human RGCs were also downregulated after the KD. OE of *KLF7* in eye progenitor cells pushed the cells towards RGC lineage. Some of the TFs of M1 modules (*EBF3*, *SREBP2* and *EBF1*) increased expression after *KLF7* OE. However, further studies on the mechanism of *KLF7* to drive RGC lineage could be achieved by multimodal single-cell RNAseq studies[63], which probe both genome accessibility and transcriptome from a single cell.

## Methods

**Preparation of single cells from donors**. All human studies were conducted under IRB oversight. All tissue was de-identified prior to delivery to the eye bank, which is conducted under an approved IRB protocol following US HIPAA privacy law. Tissue was then sent to the Blenkinsop lab for dissection. All de-identified medical information given was under the consent of donors. Single-cell RNAseq transcripts were predicted by poly-A tag sequencing of the first 92 bases and therefore no personal genetic information was collected that may be identifiable or traced back to donors.

Post-mortem human adult eyes were obtained from donors ageing between 28 and 84 years old within 24 h after death from the Eye Bank for Sight Restoration, New York, NY, USA and approved for research purposes. The eyes were dissected to separate individual tissue parts paying close attention not to include regions that rest at the interface between two tissues. Tissues were dissociated by 2% collagenase (Trypsin, Worthington, NJ), 3 μg/ml DNase I Solution (STEMCELL Technologies) and 2 μM Thiazovivin ROCK Inhibitor for a minimum of 2 h. Cells were filtered through a 40 μM nylon mesh membrane to remove clumps of cells. Cells were then frozen gradually in CS2 medium (Stem Cell Technologies, USA) and then stored in liquid nitrogen. Cells were shipped in dry ice for transport. Cells were thawed and processed for single-cell analysis using 10× genomics single-cell RNAseq protocol. The single-cell library was sequenced using Hiseq 4000 Illumina sequencer. Detailed methods for dissection of several tissues of the eye compartment are kept in Supplementary Information.

**RNA FISH in human retinal slides**. Frozen Retinal Slides were obtained from Amsbio. Custom Stellaris® FISH Probes were designed against gene targets by utilising the Stellaris® RNA FISH Probe Designer (Biosearch Technologies, Inc., Petaluma, CA) available online at www.biosearchtech.com/stellarisdesigner (Version 3). The targets were hybridised with the Quasar 670 dye. Stellaris RNA FISH Probe set labelled with (Biosearch Technologies, Inc.), following the manufacturer's instructions available online at www.biosearchtech.com/stellarisprotocols. The images were taken in Zeiss AxioImager Z1 (EBL) and then processed using ImageJ software (version 1.53c). For immunoFISH Anti-PKCA(H-7) Sc-8393 (Santa Cruz Biotechnology) was used at a dilution of 1:100 and Anti-Vimentin Ab24525 (Abcam) was used at a dilution of 1:500.

**Reporting summary**. Further information on research design is available in the Nature Research Reporting Summary linked to this article.

## Data availability

Single-cell RNA-seq data (human and pig eyes) have been deposited in the Gene Expression Omnibus (GEO) under the accession code "GSE147979". Previously published single-cell RNA-seq data that were reanalysed here are available in the GEO or ArrayExpress under the accession codes "GSE118480" (macaque retina cells), "GSE63472" (mouse retina cells), "GSE135406" (zebrafish retina cells), "GSE137537" (human retina cells), and "E-MTAB-7316" (human macula/periphery cells). scRNA-seq data can be queried interactively at the Single-Cell Portal (SCP) under the accession code "SCP1311" (primary tissues) and "SCP1386", "SCP1387", "SCP1388", "SCP1389", "SCP1390" and "SCP1391" (in vitro RGC differentiation). All other relevant data supporting the key findings of this study are available within the article and its Supplementary Information files or from the corresponding author upon reasonable request. Source data are provided with this paper.

## Code availability

The source codes used in this study can be found in Supplemental Data. Unless mentioned otherwise, all plots were generated using the R package "ggplot2" and "pheatmap". The proposed models and schematic workflow were illustrated using BioRender (https://biorender.com).

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

## Acknowledgements

We thank the donors and their families for whom this work could not have been possible. We are grateful to Chadi El Farran and Haofei Wang for their helpful contributions to productive discussions. We also thank Vanessa Lorraine Chea for her help in editing the manuscript. We also thank the Spatial and Single Cell Genomics Platform at Genome Institute of Singapore for providing a single-cell RNA sequencing facility. We are thankful to Central Imaging Facility at IMCB for providing microscope resources. HLL is supported by the Glenn Foundation for Medical Research, the Mayo Clinic Centre for Biomedical Discovery and the Mayo Clinic Cancer Centre. Y.-H.L. is supported by the [NRF Investigatorship award—NRFI2018-02] and [JCO Development Programme Grant—1534n00153] grants. The work was supported by NIH/NEI grants EY029736 and EY030215. Finally, we are grateful to the Biomedical Research Council, Agency for Science, Technology and Research, Singapore, for research funding.

## Author contributions

P.G. performed experiments and wrote the paper. K.H. performed most bioinformatics analyses and wrote the paper; Y.C., Y.Y.Z. and B.M. contributed equally to the paper. Y.C. performed experiments and assisted with manuscript writing. Y.Y.Z. performed a part of bioinformatics analysis. B.M., B.H.P. and L.H.Y. assisted in performing experiments. K.A.L. assisted with the paper writing; X.S., R.C.B.W., K.W.C., H.L. and T.A.B. analysed the data; and Y.H.L. designed the study, analysed the data and wrote the paper.

## Competing interests

The authors declare no competing interests.
