## [Peer Review File · Nature Communications]

Reviewers' Comments:

Reviewer #1:

Remarks to the Author:

This manuscript describes an interesting resource in terms of single cell sequencing of retina, cornea, choroid and ciliary body in two species (human and pig). Despite several high quality single cell datasets exist for the human retina, this manuscript provides sequencing of additional tissues as well as in the pig eye. My main concern is that apart from the value in the data provided to the community by this work, there are no new insights supported by experimental evidence. The work is very descriptive and also somewhat superficial, for example despite the sequencing of multiple species, not much is gained from their comparison and the zebrafish data are not commented upon at all. Also the authors make interesting hypotheses on a huge range of topics ranging from the discovery of distinct population of putative adult stem cells, that by itself would be a major discovery, to viral entry points and immune response to viral infection, none of them however supported by experiments, but only based on expressed genes. Hence the manuscript is very speculative, and this is a pity as if the authors decided to follow up on one of their hypotheses, the manuscript would be much much stronger and interesting.

Finally, the SARS COV 2 in the title is really out of context and all the discussion on the expression of genes related to viral infection does not really add much to the research in this field, also because this seems not to be a major route of infection in humans.

Reviewer #2:

Remarks to the Author:

In this manuscript, the authors have provided a single cell transcriptomic atlas of human eye as a resource. Different from previous studies, the authors paid particular attention to the other cell types in addition to neural retina cells. Though the majority of the work is purely descriptive, it strengthens the single cell atlas of human eyes. It would be better if the authors can link their dataset to a typical eye disease. SARS-CoV-2 can infect human eyes is well known and to examine SARS-CoV-2 receptors expression in single cell data has been repeatedly done.

Reviewer #3:

Remarks to the Author:

The authors performed scRNA-Seq on human ocular compartments of four donor eyes. A total of 33,445 cells are profiled. By examining the expression profile of genes between human, porcine, and zebrafish, the authors show compared inter-species transcriptomic expression data. Further the authors show a map of receptors specific to SARS-CoV-2 within human ocular cells. This is a well-designed study that would be significant resource to the community and I feel would influence thinking in the field. I would accept the manuscript though would ensure that the data and source code are all readily available and accessible for the community to use.

- It is essential for single cell data to be made available in an easily browsable/explorable way (there are several platforms from the community that the authors could use to achieve this quickly). In the manuscript, the authors should mention this aspect and include source code in the methods. The authors also need to make the data readily available in an easily browsable or explorable way to make it useful for the community to use so that a reader can easily check or explore a single gene's expression in a cell-type. An example is the Broad Single Cell Portal though there are other similar ones, https://singlecell.broadinstitute.org/single_cell/study/SCP839/cell-atlas-of-the-human-fovea-and-peripheral-retina

- The authors need to write out what the abbreviation stands for. For example I haven't heard the term RIVI used before, and it takes time for the readers to need to look back the manuscript to find out what it stands for. Similarly for NCs, the authors should write out neuroblasts as NC and activated T cells as such and not ATC as NC also refers to neurocrest. CM should be written as monocytes, FB as fibroblasts as CF are also referred to as fibroblasts. I have not heard of the

terms CCPAI, DCEC, POSC, and the authors should write out fully all the jargon terminology and cell-types to make the paper easier to follow.

- The authors should provide more details on the donors including ie were they all healthy, or did any of the four have any ocular diseases? What is the postmortem interval, gender, and age of each of them? The authors should provide specific details in the methods as to the regions collected (ie what region of the retina, macular or periphery was sequenced? Was the entire cornea or limbus where corneal stem cells collected? What was the viability of the cells following thawing the cells after freezing? Were nonviable cells sequenced or just the viable ones following thawing? Were cells all mixed from different compartments of the eye or each compartment sequenced individually?

- The authors state that they identify novel markers such as RAB41 in cone photoreceptors in humans; however, in figure 3D of (Armamoto et al., FIN-Seq: transcriptional profiling of specific cell types from frozen archived tissue of the human central nervous system) the authors identified RAB41 as specific for cone photoreceptors in humans. The authors should reference this work.

- The FISH images shown in Fig. 2 are challenging to interpret. Could the authors use a technique used for validation that has at least cellular resolution and therefore be able to assign the expression to a particular cell (e.g., immune histochemistry, morphological, FACS-based). A validation by FISH like in figure 2 may not be sufficient since the cell crowding does not allow a proper evaluation of colocalization in a cell.

- The authors claim that NIF3L1 localizes in inner nuclear layer, which consists of Rod BP; however, INL has many other cell types (MG, AC, HC, cone bipolar cells). Can the authors colocalize NIF3L1 using immunohistochemistry for a rod bipolar marker such as PKCalpha to make the claim that it is a novel marker. The authors appear to be pointing to co-localized RHO rods in the INL; however, rods are not located in the INL.

- Can the authors label the colors in suppl Fig 2F as it is challenging to tell the cell types apart? The text states that horizontal cells have the highest G2M gene expression; however, can the authors elaborate on this as HC are not typically undergoing cell division in humans?

- The authors recent manuscript on Biorxiv (Potential modes of COVID-19 transmission from human eye revealed by single-cell atlas, Hamashima et al., Biorxiv) appears to have a similar conclusion regarding Figure 1 of (Hamashima et al., Biorxiv, Potential modes of COVID-19 transmission from human eye revealed by single-cell atlas) and figure 7A from (Single-cell reconstruction of the regulons in the human ocular compartments and their susceptibility to SARS-CoV-2) with the cell-types in the human eye that express ACE2 and are susceptible to SARS-CoV-2 infection. I would ensure that the Biorxiv manuscript references the manuscript submitted to Nature Comm.

REVIEWER 1

GENERAL COMMENTS

This manuscript describes an interesting resource in terms of single cell sequencing of retina, cornea, choroid, and ciliary body in two species (human and pig). Despite several high-quality single cell dataset exists for the human retina, this manuscript provides sequencing of additional tissues as well as in the pig eye. My main concern is that apart from the value in the data provided to the community by this work, there are no new insights supported by experimental evidence.

We are grateful to the reviewer for the comment. To answer the comment of no new insights supported by experimental evidence, we have included KLF7 knockdown and overexpression experiments to support role of KLF7 in retinal ganglion cell genesis.

SPECIFIC COMMENTS

The work is very descriptive and also somewhat superficial, for example despite the sequencing of multiple species, not much is gained from their comparison and the zebrafish data are not commented upon at all.

We are grateful to the reviewer for the comment. We have worked towards understanding conserved gene patterns across species in retinal cell-types to answer this comment in Fig R1 (Fig 2f – Fig 2h of revised manuscript). Besides that, we also investigated conserved gene patterns across human and pigs in iris cell-types. We have identified patterns specific to human cell-types in retina using single cell Coordinated Gene Association in Pattern Sets (scCoGAPS) algorithms. These patterns are unique to cell-types. For example, pattern 18,17,39 and 42 (shown by green circles) are specific to Muller glial cells in humans.

Fig R1: Patterns of gene expression as determined by scCoGAPS algorithm in retinal cell-types of human eye. Circled region shows pattern specific to Muller glial cells.

The patterns specific to cell-types in human retina were checked across mouse, pig, zebrafish and macaque retinal cell-types. Such patterns were checked across species (Fig R2, SFig 2d-2k in updated manuscript). Upon checking, we found out that while some patterns were conserved across the species in cell-type and some are not. This gives us resources to explore role of genes which are species-specific and genes that are common across species. To give an example, we focussed on the common pattern across species in Muller glial cells (Fig R3, Fig 2g in updated manuscript).

Fig R2: The patterns of gene expression in human retinal cells projected into cross-species retinal cell-types.

Fig R3: Circled red region shows that pattern 17 shows high expression in Muller glial cells across species.

We found out conserved pattern of gene expression like Pattern 17 across species in Muller glial cells (Fig R3, Fig 2g in updated manuscript). To explore further what constitutes Pattern 17, we expanded upon the genes of it (Fig R4, Fig 2h in updated manuscript).

Fig R4: Red selected region shows the genes that belong to Pattern 17 and are specific to Muller glial cells in humans. Blue rectangles shows interesting genes that are important in Muller glial cells.

The genes that make comprise pattern 17 are well known markers of Muller glial cells. Gwon et al. showed that CLU gene was expressed in the INL layer of normal rat retina. Grosche et al. showed the GLUL transcripts was enriched in Muller glial cell factions during MACS sorting of retinal cells. CRABP1 was shown to upregulate in muller glial cells of avians during retinal damage by Todd et al. Greenberg et al. have shown that VIM gene promoter is one the essential promoters for Muller glial cells in rats. Morshedian et al. report that RGR opsin in Müller cells functions to regenerate cone visual pigments during light exposure in mice. Vardiman et al. have shown that CA2 gene expression becomes restricted to Muller glial cells in later part of development in chick retina. GPM6B has been shown to be specific to glial cells in brain by Choi et al. However, its role in Muller glial cells has not been reported. HES1 was reported to drive early neuron progenitors toward muller glial cell lineage by Furukawa et al. ITM2B was shown to be present in retinal progenitor cells and Muller glial cells in macular region during development by Lu et al. Inclusion of such genes in Pattern 17 shows that such genes are indispensable for Muller glial function.

Fig R5: Cluster 17 which is a conserved pattern across all species is similar to the Resting Muller glial cells in Muller glial cells of zebrafish.

We wanted to compare the cross species conserved pattern 17 in zebrafish before and after injury using the data obtained from Brodie-Kommit et al. It seems like most the correlation with pattern 17 is with resting Muller glial cells. However, the rest of the genes that are activated after injury does not share correlation with pattern 17. As the time of injury increases, the correlation of pattern 17 decreases with regenerating Muller glial cells. We checked the patterns of Muller glial cells that are activated after injury and found out they include the GO terms like “regeneration”. We also showed that Pattern 17 has very little in common with Pattern 78 which is one of the patterns that are activated after injury in Muller glial cells in zebrafish (Fig R5, Fig 2i-l in updated manuscript). This shows that the conserved module 17 across species has nothing in common with the modules related to regeneration in zebrafish and suggests it might be involved in housekeeping of Muller glial function.

Fig R6: Checking conservation of TFs by plotting pairwise correlation across species.

We describe module of TFs that populate a particular cell-type in retina. Here, we see the conservation of TF across species. For example, LHX9 is a TF that is conserved across species in amacrine cell-types. Balasubramanian et al. showed that LHX9 is required for the development of Retinal Nitric Oxide-Synthesizing Amacrine Cell Subtype (NOAC) (Fig R6, SFig 7f in the updated manuscript). One of the TFs that we have focussed on our studies, KLF7, is conserved across human, macaque, mice, and zebrafish. Besides cross-species conservation of genes and gene patterns in retina, we also focussed in the conservation of gene patterns in the iris cell-types in humans and pigs (Fig R7, Fig 4 d,e in updated manuscript).

Fig R7: Patterns of gene expression as determined by scCoGAPS algorithm in iris cell-types of human eye and projection of those patterns into pig iris cell-types.

We found out that certain pattern of genes is specific to respective cell-types across species. Like pattern 86 and 56 are specific to COL9A1 high cells of ciliary body, while patterns 87 and 13 were specific to putative stem cells. GO enrichment terms for four patterns mentioned above show terms like “Stem cell proliferation”, “neural crest differentiation” and “embryonic morphogenesis” (Fig R8, Fig 4f in updated manuscript). scCoGAPS gene pattern analysis points more evidence towards the interesting nature of Putative stem cells.

Fig R8: GO enrichment terms for patterns specific to COL9A1 high ciliary body cells and putative stem cells. Red circles show interesting GO terms.

Also the authors make interesting hypotheses on a huge range of topics ranging from the discovery of distinct population of putative adult stem cells, that by itself would be a major discovery, to viral entry points and immune response to viral infection, none of them however supported by experiments, but only based on expressed genes. Hence the manuscript is very speculative, and this is a pity as if the authors decided to follow up on one of their hypotheses, the manuscript would be much much stronger and interesting.

We are grateful to the reviewer for the comment. To answer the concerns about analysis being supported only by expressed genes and manuscript being speculative, we have focussed our efforts on understanding transcription factor regulons in retinal ganglion cells. We differentiated H9 cells to RGC cells and carried out overexpression and knockdown experiments to show the role of KLF7 in RGC genesis.

Fig R9: KLF7 knockdown causes downregulation of RGC specific module genes in differentiating RGC cells.

The knockdown was performed during RGC differentiation before maturation of RGCs (Fig R1, Fig 7e in updated manuscript). As a result of knockdown of KLF7, we can see drastic differences in the proportion of cells that are destined to be RGCs. Also, during knockdown genes like POU4F1, IRX2 and EBF3 which comprise the transcription factor modules of RGC (M1 module) got downregulated. POU4F1 is required for the early development of Retinal ganglion cells as shown by Muzyka et al. EBF3 is one of the markers of RGC development and maturation used by Brodie-Kommit et al. In RGC precursor cells, EBF1 is downregulated after knockdown, which is also one of the transcription factors included in M1 modules (Fig

R9, Fig 7c, d, f, j in updated manuscript). We also performed KLF7 overexpression in differentiating RGCs and check the differences it made after.

Fig R10: KLF7 overexpression in differentiating RGC cells increases the proportion of RGC cells and increases the expression of transcription factor expression specific to RGC module.

We overexpressed KLF7 ORF in differentiating RGC cells and found that it increased the proportion of RGC like cells (Fig R10, Fig 7h in updated manuscript). Overexpression of KLF7 also increased the overexpression of transcription factor that are specific to RGC modules like EBF3, SREBF2 and EBF1 in RGC precursor cells. Gene expression of EBF3, a marker for maturation and development of RGC, after KLF7 OE shows how KLF7 is driving other transcription factors to push cells towards RGC lineage (Fig R10, Fig 7g, i, k in updated manuscript).

Finally, the SARS COV 2 in the title is really out of context and all the discussion on the expression of genes related to viral infection does not really add much to the research in this field, also because this seems not to be a major route of infection in humans.

We are grateful to the reviewer for the comment. We have removed SARS-CoV-2 from the title and removed the portion describing expression of genes related to viral infection. Instead we have expanded on creating a disease map (Fig R11, Fig 5a,b in updated manuscript) and describing receptors related to viral entry in human eye (SFig 5b,c in updated manuscript).

In this manuscript, the authors have provided a single cell transcriptomic atlas of human eye as a resource. Different from previous studies, the authors paid particular attention to the other cell-types in addition to neural retina cells. Though the majority of the work is purely descriptive, it strengthens the single cell atlas of human eyes.

I would like to thank the reviewer for encouraging comments on our manuscript. Please find responses to specific comments.

It would be better if the authors can link their dataset to a typical eye disease. SARS-CoV-2 can infect human eyes is well known and to examine SARS-CoV-2 receptors expression in single cell data has been repeatedly done.

We are grateful to the reviewer for the comment. To address these concerns, we have focussed on creating a disease map across the different cell-type of eye in Fig R11 (Fig 5b in the updated manuscript). We have created a bubble plot for genes which mutations are involved in different conditions. To streamline, we subdivided disorders into 8 groups which are colour blindness, corneal disorders, eye cancer, eye movement disorders, macular degeneration, optic nerve disorders, retinal disorders, vision impairment, and blindness. We found out that genes that are involved in corneal disorders are also expressed in ciliary body region of eye. These genes are also expressed in the fibroblasts which provide the structural integrity to eye (Fig 5b in updated manuscript).

Fig R11: Bubble plot of genes involved in different eye disorders across cell-type of eye. Circled region shows that genes that are involved in corneal disorders are expressed in ciliary body regions also.

The authors performed scRNA-Seq on human ocular compartments of four donor eyes. A total of 33,445 cells are profiled. By examining the expression profile of genes between human, porcine, and zebrafish, the authors show compared inter-species transcriptomic expression data. Further the authors show a map of receptors specific to SARS-CoV-2 within human ocular cells. This is a well-designed study that would be significant resource to the community and I feel would influence thinking in the field.

I would like to thank the reviewer for encouraging comments on our manuscript. Please find responses to specific comments.

SPECIFIC COMMENTS

I would accept the manuscript though would ensure that the data and source code are all readily available and accessible for the community to use.

It is essential for single cell data to be made available in an easily browsable/explorable way (there are several platforms from the community that the authors could use to achieve this quickly). In the manuscript, the authors should mention this aspect and include source code in the methods. The authors also need to make the data readily available in an easily browsable or explorable way to make it useful for the community to use so that a reader can easily check or explore a single gene's expression in a cell-type. An example is the Broad Single Cell Portal though there are other similar ones, https://singlecell.broadinstitute.org/single_cell/study/SCP839/cell-atlas-of-the-human-fovea-and-peripheral-retina

To address reviewer concerns, we have uploaded our entire single cell RNAseq data to Broad single cell portal. Here are some of the screenshots of the data in the portal.

Fig R12: tSNE plot of whole eye atlas as visualized from Broad single cell plot viewer showing different subtypes of cells in different colours.

Fig R13: tSNE plot of human retina as visualized from Broad single cell plot viewer showing different subtypes of cells in different colors.

Fig R14: Violin plot of expression of ONECUT2 gene across different cell-types of retina.

Fig R15: Expression of RPE65 gene in tSNE plot of eye atlas.

The authors need to write out what the abbreviation stands for. For example, I haven't heard the term RIVI used before, and it takes time for the readers to need to look back the manuscript to find out what it stands for. Similarly for NCs, the authors should write out neuroblasts as NC and activated T cells as such and not ATC as NC also refers to neurocrest. CM should be written as monocytes, FB as fibroblasts as CF are also referred to as fibroblasts. I have not heard of the terms CCPAI, DCEC, POSC, and the authors should write out fully all the jargon terminology and cell-types to make the paper easier to follow.

We appreciate the comments from the reviewer. We have changed the names of cell-types so that it is easier for readers to follow. RIVI has been changed to COL9A1-high ciliary body cells. Neuroblasts have been labelled as Pigmented ciliary body cells. The suggestions to label cells without jargon has been followed in the manuscript. Here is the table of their old names and new names of cell-types.

Old cell annotation	New cell annotation
Regeneration Incompetent cells of Iris (RIVI)	COL9A1-high ciliary body cells
Neuroblast cells (NC)	Pigmented ciliary body cells.
CD14+ monocytes (CM)	Monocytes
Differentiating Corneal Epithelial Cells (DCEC)	ELF3- high corneal epithelial cells
Corneal cells proliferating after differentiation (CCPAI)	TGFBI- high corneal epithelial cells
Crystallin Producing cells (CPC)	CRYAA- high ciliary body cells
Iris Stem/Progenitor cells (ISPC)	Putative stem cells
Melanin+ cells (MCs)	Melanocytes
Schwann cells of Iris Stroma (SCOIS)	Schwann cells

Conjunctival Epithelial Cells (CJEC)	Conjunctival cells
Limbal Melanocytes (LM)	Melanocytes
MEG3+ Fibroblasts (MGF)	MEG3-high fibroblast cells
WIF1+ Fibroblasts (WF)	WIF1- high fibroblast cells

The authors should provide more details on the donors including ie were they all healthy, or did any of the four have any ocular diseases? What is the postmortem interval, gender, and age of each of them?

We appreciate the comments from the reviewer. The information about donor, postmortem interval, health complications, gender and age has been included in Supplementary Fig 1j. The donors died of cardiac, respiratory, or renal failure with no ocular disease history.

Donor	Sex	Age	Retrieval Time	Complications
72	Female	84	3 hr 18 min	Respiratory failure
73	Female	86	3 hr 58 min	Rectal Cancer
74	Female	28	7 hr 48 min	None
75	Male	39	2 hr 03 min	Astrocytoma
76	Male	84	3 hr	None
78	Female	81	5 hr 20 min	Lung Cancer

The authors should provide specific details in the methods as to the regions collected (ie what region of the retina, macular or periphery was sequenced?)

We appreciate the comments from the reviewer. The specific details of the method are included in the revised manuscript.

ISOLATION OF CORNEA

The anterior segment was separated from the rest of the eye by performing a circumferential incision 6mm posterior from the ora serrata. The lens, ciliary body and iris were then removed by manual manipulation using forceps. Conjunctiva was carefully dissected off sclera. Cornea cells were isolated from the central part of the cornea leaving approximately 1mm distance from the edge of the transparent section of the anterior segment of the eye. This corneal tissue was then further cut into 1mm² pieces and digested with collagenase 1mg/ml. The full thickness was used including corneal epithelium, corneal stroma, and corneal endothelium. The limbal region was dissected from the remaining tissue after the cornea was removed. The limbus sample included the full thickness of 1mm of transparent cornea out to 1mm of opaque tissue.

ISOLATION OF SCLERA

The sclera sample included the full thickness of greater than 2mm away from the beginning of opaque region of the anterior segment to 6mm of opaque region.

ISOLATION OF CILIARY BODY AND IRIS

The ciliary body and iris were pulled away from the cornea/limbus/sclera tissue using #5 Dumont forceps. The iris was then gently pulled away from the ciliary body.

These separated tissues of sclera, ciliary body and iris cells were then further cut into 1mm² pieces and digested with collagenase 1mg/ml with 3ug/ml DNase Solution (STEMCELL Technologies) in Earle's balanced salt solution for 3 hours. Following digestion, tissue pieces were triturated for 2 minutes using a 10ml pipette. The tissue pieces are allowed to settle at the bottom of the conical tube and the supernatant is collected, centrifuged, and resuspended and frozen using CS₂ medium (Cryostore).

ISOLATION OF RETINA

Taking the posterior segment, the vitreous and retina was encouraged with the use of angled forceps to separate from the back of the eyecup to the point where the retina was only still attached to the eye at the location of the optic disk. Using micro scissors the retina was cut from the optic disk. The vitreous was then separated from the retina using forceps and the whole retina was cut into approximately 1mm² pieces and digested with hyaluronidase (6mg/ml) in Earle's balanced salt solution for 3 hours. The tissue pieces are allowed to settle at the bottom of the conical tube and the supernatant is collected, centrifuged, resuspended and frozen using CS₂ medium (Cryostore). The retina was processed without separating them into macular or periphery region.

ISOLATION OF RETINAL PIGMENTED EPITHELIUM AND CHOROID

The posterior eyecup was placed in a cup with the optic nerve facing down. The eyecup is filled with 0.25% trypsin with 3ug/ml DNase Solution and incubated at 37°C for 50 minutes. RPE were then brushed off the Bruch's Membrane, collected and frozen using CS₂ medium. After the RPE are removed, choriocapillaris/Bruch's membrane is cut into 1mm² pieces of tissue and placed into 2% collagenase 1mg/ml with 3ug/ml DNase Solution. Afterwards, tissue pieces are triturated for 2 minutes using a 10ml pipette. The tissue pieces are allowed to settle at the bottom of the conical tube and the supernatant is collected, centrifuged, and resuspended and frozen using CS₂ medium (Cryostore).

What was the viability of the cells following thawing the cells after freezing?

We appreciate the comments from the reviewer. The viability of cells following thawing was differed according to cell-types as checked by trypan blue staining. For example, the viability of corneal cells and iris cells was determined to be around 65 percent and 17.8 percent respectively by trypan blue staining. We also checked the viability bioinformatically by calculating the ratio of cells that were viable to the cells that were target for scRNAseq. The viability of corneal cells and iris cells ranged from 40-62 percent and 14.8-74.55 percent respectively when computed bioinformatically.

Total cell concentration: 3.49×10^5 cells/mL
 Live cell concentration: 6.22×10^4 cells/mL
 Dead cell concentration: 2.87×10^5 cells/mL
 Viability: 17.8 %
 Average cell size: 10.0 μm
 Total cell number: 73
 Live cell number: 13
 Dead cell number: 60

Total cell concentration: 3.59×10^5 cells/mL
 Live cell concentration: 2.35×10^5 cells/mL
 Dead cell concentration: 1.24×10^5 cells/mL
 Viability: 65.3 %
 Average cell size: 13.4 μm
 Total cell number: 75
 Live cell number: 49
 Dead cell number: 26

	Cornea72	Cornea73	Cornea76	Iris73	Iris74	IrisPE75	IrisStroma75	IrisPE76	IrisStroma76	IrisMuscle76
Input Cells	6000	6000	6000	6000	6000	6000	6000	6000	6000	6000
After first filtering (Cellranger) *1	4559	4921	3908	2221	3522	3665	4923	1332	1940	1198
After second filtering (Seurat) *2	3439	2426	3736	1293	1107	1812	4473	889	1278	1141
Computationally estimated variability (%)	57.2	40.43	62.2	21.55	18.4	30.6	74.55	14.81	21.3	19

Fig R16: Representative cell viability percentages of cells as checked by trypan blue staining. They range between 17.8- 65.3 percent. The samples that were thawed were iris cells and corneal cells, respectively. The table shows computational estimation of viability of cell-types. First filtering involved removing low quality/dead/barcoding-unsuccessful cells in cellranger software. Second filtering involved removing low quality cells and outliers using Seurat package.

Were nonviable cells sequenced or just the viable ones following thawing?

We appreciate the comments from the reviewer. We did not remove non-viable cell as it increased the time for scRNAseq preparation which might exacerbate health of cells. We exclude cells that have low gene counts and high mitochondrial gene expression bioinformatically. As you can see from our bioinformatic analysis, the percent of viable cells used for analysis ranges between 18-64 percent which matches with our trypan blue staining.

Fig R17: Filtration of cells by removing cells with low UMI counts and high mitochondrial cell expression in corneal cells.

Fig R18: Filtration of cells by removing cells with low UMI counts and high mitochondrial cell expression in iris cells.

Were cells all mixed from different compartments of the eye or each compartment sequenced individually?

We thank the reviewer for this comments. We dissected eye compartments into different parts as mentioned in Methods. We separated Choroid, Cornea, Iris Muscle, Iris pigmented epithelium, Iris stroma, Retina, RPE, and Sclera physically, and each of these tissues were sequenced individually.

The authors state that they identify novel markers such as RAB41 in cone photoreceptors in humans; however, in figure 3D of (Armamoto et al., FIN-Seq: transcriptional profiling of specific cell-types from frozen archived tissue of the human central nervous system) the authors identified RAB41 as specific for cone photoreceptors in humans. The authors should reference this work.

We appreciate the comments from the reviewer. We have included the reference from Armamoto et al. in our updated manuscript.

The FISH images shown in Fig. 2 are challenging to interpret. Could the authors use a technique used for validation that has at least cellular resolution and therefore be able to assign the expression to a particular cell (e.g., immune histochemistry, morphological, FACS-based). A validation by FISH like in figure 2 may not be sufficient since the cell crowding does not allow a proper evaluation of colocalization in a cell.

We appreciate the comments from the reviewer. We have used immune-FISH to colocalize some of RNA FISH probes with PKCalpha and VIM antibodies. Also, we have added MALAT1 as an internal control for RNA FISH so that it could be easier to interpret the data (Fig R19, Fig 2c-e in updated manuscript). Besides that, we have also performed

immunofluorescence for TFs to show colocalization (Fig7b, SFig 7a in updated manuscript). The RNA FISH spots have been circled in the figure.

Fig R19: RNA FISH localization of NIF3L1 with PKCA and TRDN with VIM. MALAT1 is used as an internal control in both images. Circled red regions shows the RNA FISH spots of NIF3L1 and TRDN.

- The authors claim that NIF3L1 localizes in inner nuclear layer, which consists of Rod BP; however, INL has many other cell-types (MG, AC, HC, cone bipolar cells). Can the authors colocalize NIF3L1 using immunohistochemistry for a rod bipolar marker such as PKCalpha to make the claim that it is a novel marker. The authors appear to be pointing to co-localized RHO rods in the INL; however, rods are not located in the INL.

We appreciate the comments from the reviewer. As suggested by reviewer, we have performed PKCA localization with NIF3L1 as shown in Fig R19.

Can the authors label the colors in suppl Fig 2F as it is challenging to tell the cell-types apart?

We appreciate the comments from the reviewer. We have updated that in our updated manuscript figures.

The text states that horizontal cells have the highest G2M gene expression; however, can the authors elaborate on this as HC are not typically undergoing cell division in humans?

We thank reviewer for the comments. The cell cycle score was computed from the CellCycleScoring function in Seurat package. It uses a list of genes that were used by Scialdone et al. When we used raw counts of gene expression of cell cycles genes to create an updated PCA plot of G2M/S. Here, we see that G2/M score for horizontal cells isnt as high as before. We suspect that previous PCA plot showed high G2/M score for horizontal cells because normalized data (corrected counts) was used instead of raw counts (Fig R20).

Fig R20: Red circles shows the G2M score of horizontal cells doesn't show high G2M score after correction using raw counts to generate plot.

- The authors recent manuscript on Biorxiv (Potential modes of COVID-19 transmission from human eye revealed by single-cell atlas, Hamashima et al., Biorxiv) appears to have a similar conclusion regarding Figure 1 of (Hamashima et al., Biorxiv, Potential modes of COVID-19 transmission from human eye revealed by single-cell atlas) and figure 7A from (Single-cell reconstruction of the regulons in the human ocular compartments and their susceptibility to SARS-CoV-2) with the cell-types in the human eye that express ACE2 and are susceptible to SARS-CoV-2 infection. I would ensure that the Biorxiv manuscript references the manuscript submitted to Nature Comm.

We appreciate the comments from the reviewer. Since the bioarchive manuscript is a part of our current manuscript, we will put a link in the bioarchive paper to the current manuscript submitted to Nature communications after it is published.

1. Gwon, J. S., Kim, I. B., Lee, M. Y., Oh, S. J., & Chun, M. H. (2004). Expression of clusterin in Müller cells of the rat retina after pressure-induced ischemia. *Glia*, *47*(1), 35-45.
2. Grosche, A., Hauser, A., Lepper, M. F., Mayo, R., von Toerne, C., Merl-Pham, J., & Hauck, S. M. (2016). The proteome of native adult Müller glial cells from murine retina. *Molecular & cellular proteomics*, *15*(2), 462-480.
3. Todd, L., Suarez, L., Quinn, C., & Fischer, A. J. (2018). Retinoic Acid-Signaling Regulates the Proliferative and Neurogenic Capacity of Müller Glia-Derived Progenitor Cells in the Avian Retina. *Stem Cells*, *36*(3), 392-405.
4. Greenberg, K. P., Geller, S. F., Schaffer, D. V., & Flannery, J. G. (2007). Targeted transgene expression in muller glia of normal and diseased retinas using lentiviral vectors. *Investigative ophthalmology & visual science*, *48*(4), 1844-1852.
5. Morshedian, A., Kaylor, J. J., Ng, S. Y., Tsan, A., Frederiksen, R., Xu, T., ... & Travis, G. H. (2019). Light-driven regeneration of cone visual pigments through a mechanism involving RGR opsin in Müller glial cells. *Neuron*, *102*(6), 1172-1183.
6. Vardimon, L., Fox, L. E., & Moscona, A. A. (1986). Developmental regulation of glutamine synthetase and carbonic anhydrase II in neural retina. *Proceedings of the National Academy of Sciences*, *83*(23), 9060-9064.
7. Choi, K. M., Kim, J. Y., & Kim, Y. (2013). Distribution of the immunoreactivity for glycoprotein M6B in the neurogenic niche and reactive glia in the injury penumbra following traumatic brain injury in mice. *Experimental neurobiology*, *22*(4), 277.
8. Furukawa, T., Mukherjee, S., Bao, Z. Z., Morrow, E. M., & Cepko, C. L. (2000). *rax*, *Hes1*, and *notch1* promote the formation of Müller glia by postnatal retinal progenitor cells. *Neuron*, *26*(2), 383-394.
9. Lu, Y., Shiao, F., Yi, W., Lu, S., Wu, Q., Pearson, J. D., ... & Clark, B. S. (2020). Single-cell analysis of human retina identifies evolutionarily conserved and species-specific mechanisms controlling development. *Developmental cell*, *53*(4), 473-491.
10. Brodie-Kommit, J., Clark, B. S., Shi, Q., Shiao, F., Kim, D. W., Langel, J., ... & Hattar, S. (2021). *Atoh7*-independent specification of retinal ganglion cell identity. *Science advances*, *7*(11), eabe4983.
11. Muzyka, V. V., & Badea, T. C. (2020). Genetic Interplay Between Transcription Factor *Pou4f1/Brn3a* and Neurotrophin Receptor *Ret* In Retinal Ganglion Cell Type Specification. *bioRxiv*.
12. Lee, M. S., Wan, J., & Goldman, D. (2020). *Tgfb3* collaborates with PP2A and notch signaling pathways to inhibit retina regeneration. *Elife*, *9*, e55137.
13. Wu, F., Kaczynski, T., Matheson, L. S., Liu, T., Wang, J., Turner, M., & Mu, X. (2020). *Zfp3611* and *Zfp3612* balances proliferation and differentiation in the developing retina. *bioRxiv*.
14. Balasubramanian, R., Bui, A., Dong, X., & Gan, L. (2018). *Lhx9* is required for the development of retinal nitric oxide-synthesizing amacrine cell subtype. *Molecular neurobiology*, *55*(4), 2922-2933.

15. Hao, H., Kim, D. S., Klocke, B., Johnson, K. R., Cui, K., Gotoh, N., ... & Swaroop, A. (2012). Transcriptional regulation of rod photoreceptor homeostasis revealed by in vivo NRL targetome analysis. *PLoS Genet*, 8(4), e1002649.
16. Scialdone, A., Natarajan, K. N., Saraiva, L. R., Proserpio, V., Teichmann, S. A., Stegle, O., ... & Buettner, F. (2015). Computational assignment of cell-cycle stage from single-cell transcriptome data. *Methods*, 85, 54-61.

Reviewers' Comments:

Reviewer #1:

Remarks to the Author:

The authors have improved their analyses and overall manuscript but I find it still quite hard to read and apart from data generation I do not see any interesting biological findings. Figure captions are not informative at all and too synthetic. Still the data generated make it a good resource for the research community but I am not convinced that the manuscript is a good fit for publication in Nature Comm. Moreover, there are many grammar errors and typos so a full editing is needed prior to publication.

Reviewer #2:

Remarks to the Author:

I appreciate the authors' efforts to improve their manuscript. Though not much insight has been learned from this work, it still serves as a good research resource to the community. Therefore I recommend its publication on Nature Communications.

Reviewer #3:

Remarks to the Author:

The authors have adequately addressed my concerns with the previous version of the manuscript and I am now supportive of publication.